# High-resolution annotation of the mouse preimplantation embryo transcriptome using long-read sequencing

Yunbo Qiao[1,8✉], Chao Ren[2,8], Shisheng Huang[3,4,8], Jie Yuan[2,8], Xingchen Liu[4,5,8], Jiao Fan[6], Jianxiang Lin[1], Susu Wu[1], Qiuzhen Chen[2,7], Xiaochen Bo[2], Xiangyang Li[3,4], Xingxu Huang[3,4], Zhen Liu[5✉] & Wenjie Shu [2✉]

The transcriptome of the preimplantation mouse embryo has been previously annotated by short-read sequencing, with limited coverage and accuracy. Here we utilize a low-cell number transcriptome based on the Smart-seq2 method to perform long-read sequencing. Our analysis describes additional novel transcripts and complexity of the preimplantation transcriptome, identifying 2280 potential novel transcripts from previously unannotated loci and 6289 novel splicing isoforms from previously annotated genes. Notably, these novel transcripts and isoforms with transcription start sites are enriched for an active promoter modification, H3K4me3. Moreover, we generate a more complete and precise transcriptome by combining long-read and short-read data during early embryogenesis. Based on this approach, we identify a previously undescribed isoform of *Kdm4dl* with a modified mRNA reading frame and a novel noncoding gene designated *XLOC_004958*. Depletion of *Kdm4dl* or *XLOC_004958* led to abnormal blastocyst development. Thus, our data provide a high-resolution and more precise transcriptome during preimplantation mouse embryogenesis.

[1] Precise Genome Engineering Center, School of Life Sciences, Guangzhou University, 510006 Guangzhou, China. [2] Department of Biotechnology, Beijing Institute of Radiation Medicine, 100850 Beijing, China. [3] School of Life Science and Technology, ShanghaiTech University, 201210 Shanghai, China. [4] University of Chinese Academy of Sciences, 100049 Beijing, China. [5] Institute of Neuroscience, Key Laboratory of Primate Neurobiology, CAS Center for Excellence in Brain Science and Intelligence Technology, Chinese Academy of Sciences, 200031 Shanghai, China. [6] Institute of Geriatrics & National Clinical Research Center of Geriatrics Disease, 2nd Medical Center of Chinese PLA General Hospital, 100853 Beijing, China. [7] Computer School, University of South China, 421001 Hengyang, China. [8] These authors contributed equally: Yunbo Qiao, Chao Ren, Shisheng Huang, Jie Yuan, Xingchen Liu. ✉email: ybqiao@gzhu.edu.cn; zliu2010@ion.ac.cn; wenjie.shu@gmail.com

Transcriptome analysis based on high-throughput short-read sequencing (RNA-seq) has greatly facilitated the annotation of functional, physiological, and biosynthetic cellular states linked to spatial-temporal transitions at the organismal, tissue, and even single-cell levels[1,2]. However, the application of transcriptomic studies is limited by the accuracy and integrity of current reference sequence annotations, which may lead to misleading or incorrect conclusions about some biological events. Preimplantation embryo development, including fertilization, cell cleavage, and morula and blastocyst formation, characterized by the rapid cleavage of embryonic cells and epigenetic reprogramming of DNA methylomes, histone modifications, and chromatin structures[3–5], is the basis for healthy reproduction. In particular, the precisely controlled zygotic genome activation (ZGA) process is the foundation of the whole body plan[6]. On the other hand, the decay of maternal inherited RNAs and the activation of the embryonic genome in the ZGA process are highly correlated with parent-of-origin effects, particularly epigenetic features-dependent parental allelic expression, in transcriptome activation and epigenome reprogramming[7,8]. To understand preimplantation embryo development, we and others have defined the transcriptomes of early mouse, human, and zebrafish embryos, and identified thousands of ZGA-related genes[4,5,9,10]. In addition, our recent work showed that a disorder of methylation asymmetry in zygotes could lead to defects in embryo development[11]. Despite significant progress in this field, the use of short-read sequencing as the principal technology for the transcriptome annotation of preimplantation embryos[4,5,7], fails to resolve the complete transcriptome and the transcriptional complexity associated with novel transcripts and alternative splicing events, which occur ubiquitously in eukaryotic organisms, due to the sequencing length limitations of this method[12]. For instance, the transcriptome profiling of embryos using RNA-seq has revealed that a large number of transcripts are missing from the reference annotations[9,13]. In addition, many genome assemblies still include misassembled genomes and genomic deletions, which may lead to misassembled transcriptomes[14].

Accordingly, long-read sequencing on the Pacific Biosciences platform, representing the third generation of DNA sequencing technology, can overcome the limitations of RNA-seq via the direct reading of full-length transcripts with technologically and computationally improved sequencing error rates[15]. Long-read sequencing has become a powerful tool for defining novel transcribed regions and novel splicing isoforms, further improving knowledge of the transcriptomes of model organisms and human diseases[16–19]. Although hundreds of ZGA-related genes have been reported, the functional regulators involved in the ZGA process remain largely unclear. Even Dux, a previously well-accepted ZGA factor[20], has recently been proposed to be a dispensable gene for this process[21]. These findings drove us to identify novel transcripts and splicing variants involved in ZGA and preimplantation embryogenesis. The power of long-read sequencing also helps to avoid the large amount of technical noise derived from single-cell RNA-seq in the detection of allele-specific transcripts.

In the present study, we utilize the single-molecule real-time (SMRT) platform of Pacific Biosciences for full-length sequencing with high accuracy to improve the annotation of the mouse preimplantation transcriptome and generate a more accurately described transcriptome with additional novel transcripts and splicing isoforms. We also discover two novel transcripts that are functionally involved in early embryogenesis. Thus, our study provides an important resource for the high-resolution annotation of the transcriptome for interpreting early embryo development.

## Results

**High-resolution annotation of the preimplantation mouse embryo transcriptome.** To characterize preimplantation embryo transcripts with more accurate information, we profiled the transcriptome using both long-read and short-read sequencing across mouse preimplantation developmental embryos derived from C57BL/6J (female) × DBA/2 (male) crosses (Fig. 1a). Two batches of samples at each stage were collected. Pooled embryos at each stage from one batch, including 150 oocytes (Oo), 150 1-cell embryos (1C), 100 2-cell embryos (2C), 50 4-cell embryos (4C), 25 8-cell embryos (8C), 20 blastocysts (BL, 32–64C), and bulk sperms, were collected for cell lysis. For each stage, total RNA samples were precipitated with absolute ethanol, and the dissolved RNAs were reverse-transcribed and amplified[22]. Then, the amplified cDNAs were purified. Two batches of cDNAs at each stage were mixed together in equal amounts to obtain a sufficient amount of cDNAs (up to 2 μg) for library construction and subsequent sequencing on the Pacific Biosciences (PacBio) SMRT platform. To compare with long-read sequencing parallelly, a small fraction of the purified cDNAs of each batch was also subjected to short-read sequencing (Fig. 1a). High-quality data were generally obtained from both short- and long-read sequencing (Supplementary Fig. 1a, b).

In total, 7034–8831 high-quality non-redundant long-read transcripts were identified in the seven stages, and a total of 22,612 merged transcripts were obtained (Fig. 1b, c). We compared the length and similarity of the long-read transcripts with the GENCODE annotation (vM20) (Supplementary Fig. 1c), showing that the distribution of long-read transcripts was greater than in previously annotated references. Then, we analyzed the long-read transcripts for similarity in exon–intron structures with the annotated transcriptome. The proportion of annotated transcripts containing transcripts that were exactly matched to the annotation (defined as exact match to annotation) or transcripts contained within the GENCODE annotation (reference-annotated exonic subsets of transcripts with missing exons were defined as a sequential subset of exons contained within annotation) was over 60% in each stage (Fig. 1b). Novel transcripts, including potentially novel genes (no overlapping with the reference annotation) and potentially novel isoforms, accounted for an average of 22% of the identified transcripts in each stage (Fig. 1b). We also identified ~7% of "other" transcripts, including some chromosome-crossed and genome-unrelated sequences, that could not be classified by the current software and might have been generated during the sample preparation or sequencing process, which were excluded from subsequent analysis. Then, the long-read transcripts obtained across the seven stages were merged together, and overlapping transcripts across stages were identified, revealing 2280 transcripts as potential novel genes (10.1%) and 6289 transcripts (27.8%) that were potential novel isoforms (Fig. 1c; novel transcripts were numbered and listed in Supplementary Data 1). Interestingly, the length and exon number of the novel transcripts presented a similar distribution to the annotated transcripts (Supplementary Fig. 2a, b), validating the reliability of these novel transcripts.

Next, the merged transcripts, including the annotated transcripts (transcripts that were exact match to annotation and sequential subset of exons of transcripts that were contained within annotation) and the novel transcripts (potentially novel isoforms and novel genes), were further classified into protein-coding transcripts, long non-coding RNAs (lncRNAs), and else (not belonging to protein-coding RNAs or lncRNAs, such as processed pseudogenes, snoRNAs, and rRNAs) according to the GENCODE annotation or protein-coding potential of the novel transcripts using CPAT (Coding-Potential Assessment Tool, v2.2.0)[23] (Fig. 1d, e). We found that the majority of the annotated

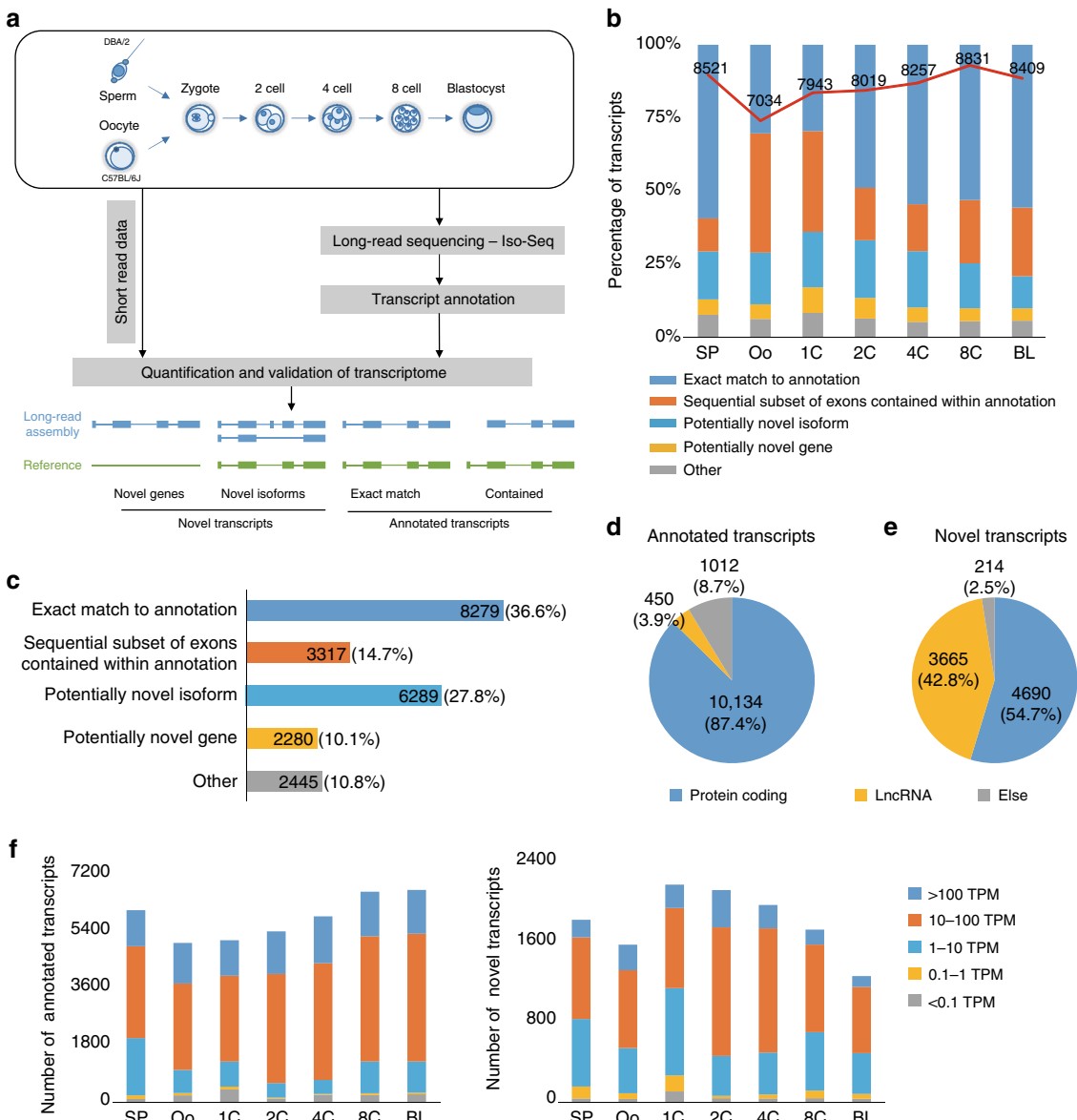

**Fig. 1 Identification of novel transcripts using PacBio SMRT sequencing in seven stages of preimplantation mouse embryos. a** Workflow for transcriptome reconstruction based on PacBio SMRT sequencing data. The Iso-seq3 pipeline was used to assemble transcripts from long-read data, and these transcripts were then mapped to the reference genome with GMAP and compared with the GENCODE (vM20) annotation using Cuffcompare. Zygotes and in vitro cultured embryos from female C57BL/6J and male DBA/2 inbred mice were collected. For a batch of samples, 150 oocytes (Oo), 150 1-cell embryos (1C), 100 2-cell embryos (2C), 50 4-cell embryos (4C), 25 8-cell embryos (8C), 20 blastocysts (BL, 32-64C), and bulk sperms, were collected for experiments. Long-read transcripts were also validated and compared to short-read data. **b** Annotation of identified long-read transcripts in the seven stages. By comparison with the GENCODE annotation, the transcripts for each stage were divided into the five indicated categories, and the percentages of transcripts in each category are shown. The red line represents the total number of transcripts in each stage. **c** Annotation of merged long-read transcripts that were the combination of transcripts identified across seven stages. The numbers and percentages of merged transcripts in the five categories are presented. **d, e** Classification of annotated (**d**) and novel (**e**) merged transcripts according to the GENCODE annotation or protein-coding potential and the length of transcripts. **f** The expression of transcripts identified from long-read data in the seven stages quantified by using short-read data. The bar plot presents the number of annotated (left) and novel (right) transcripts classified by TPM.

transcripts (10,134, 87.4%) were protein-coding transcripts, and only 450 transcripts (3.9%) were lncRNAs. In contrast, only 54.7% of novel transcripts (4690) were defined as protein-coding transcripts, and nearly half of the novel transcripts (3665, 42.8%) were classified as lncRNAs (Fig. 1d, e). Moreover, the proportions of the three categories (protein, lncRNA, and else) among both the annotated and novel transcripts in the seven stages were generally consistent with those in the merged transcripts (Fig. 1d, e; Supplementary Fig. 2c, d). Taking oocyte for an example, we

compared our merged transcriptome for oocyte stage with a published oocyte transcriptome by short-read sequencing[24]. It showed that the number and length of exons in proportion from our merged transcripts were comparable with the published oocyte data[24] (Supplementary Fig. 2e, f), further verifying the reliability of our augmented transcriptome.

Moreover, we compared the novel transcripts identified from long-read data and/or short-read data with the annotated transcripts from GENCODE. In general, novel transcripts

identified from the long-read data were more similar to the GENCODE annotation than those from the short-read data (Supplementary Fig. 3a–f). In addition, the transcripts that could only be identified from long-read data displayed slightly more similar characteristics to the GENCODE-annotated transcripts than the transcripts that could only be identified from short-read data (Supplementary Fig. 4a–f).

To determine whether long-read transcripts were supported by the short-read RNA-seq data, we quantified the expression levels of both annotated and novel transcripts in the seven stages with corresponding short-read sequencing data. In each stage, over 90% of the annotated and novel transcripts were identified with a TPM (transcripts per million) >1, and more than half of these transcripts were highly expressed (TPM > 10) in all stages (Fig. 1f). These results indicate that most of our identified annotated and novel transcripts are abundantly expressed in preimplantation embryos.

**Validation of long-read transcripts in mouse embryos.** To assess the ability of identifying novel transcripts with long-read sequencing data, we compared the saturation curve of the novel transcripts identified from short-read data only with that of novel transcripts derived from merging transcripts identified from short- and long-read data. We observed that the number of novel transcripts identified from the combination of short- and long-read data was much higher than that identified from only by short-read data (Fig. 2a), further demonstrating the advantages of long-read data for the identification of more potential novel transcripts. To further characterize the identified novel coding transcripts that function as putative protein-coding genes, we assessed these high-coding-potential transcripts (Fig. 1e) with a minimum ORF (length greater than 300-bp) via sequence homology analysis with the Uniprot protein database[25] using BLASTP (v2.9.0)[26] and functional protein domain prediction using *hmmer* (v3.2.1) with the Pfam database (v31.0)[27,28]. We found that the vast majority of novel coding transcripts were predicted to encode functional proteins (Fig. 2b). According to the observed sequence homology with the annotated protein information, novel coding transcripts were subjected to gene ontology (GO) enrichment analysis, showed that these novel coding transcripts had important roles in cellular, metabolic process, and biological regulation (Supplementary Fig. 5a).

Compared with the fully explored protein-coding regions of the whole genome, the functions of non-coding sequences, which constitute over 98% of the genome, remain largely unknown. Therefore, we evaluated whether these novel non-coding transcripts were evolutionarily conserved. We performed sequence conservation analysis according to both phyloP[29] and phastCons[30] scores. In general, the level of sequence conservation for novel non-coding transcripts was significantly higher than that of random control regions (Supplementary Fig. 5b, c). Specifically, 1183 transcripts (30.5%) among the novel non-coding transcripts showed higher conservation scores than those calculated from random control regions (phyloP[29] and/or phastCons[30]) (Fig. 2c), indicating the conservation of our newly identified non-coding transcripts.

To validate our identified novel transcripts, we categorized the transcriptional start sites (TSSs) of novel transcripts into three groups based on transcript classification and GENCODE annotation (see Method section). Among 6289 potentially novel isoforms, 4273 transcripts possessed classic TSSs overlapping with the TSSs of previously annotated isoforms, and 2014 novel transcripts possessed potentially novel TSSs; among the potentially novel genes, 2266 transcripts might have potentially novel TSSs (Supplementary Data 2). To confirm this possibility, we

profiled the histone H3 trimethylation (H3K4me3) enrichment, which is a chromatin signature of active promoters, around these related TSS regions in preimplantation embryos[31]. As expected, H3K4me3 showed significant enrichment in many promoters of three groups of TSSs (Supplementary Fig. 6a) that was similar to the bimodal distributions of H3K4me3 in the promoters of annotated TSSs (Supplementary Fig. 6b–d). Generally, the H3K4me3 signal intensity was roughly correlated with gene expression levels, although some genes with low expression levels were also enriched with H3K4me3 signals because the basal modification levels were distinct for different genes (Supplementary Fig. 6a).

Because H3K4me3 signal intensity in the novel TSSs for some potentially novel isoforms and novel genes was weak, we tried to identify high-confidence novel TSSs for potentially novel isoforms and novel genes. Thus, we enriched the TSSs that overlapped with significant H3K4me3 peaks (500-bp length around the TSSs) and obtained 1200 (60.0%) and 1007 (44.4%) high-confidence novel TSSs for potentially novel isoforms and novel genes, respectively (Fig. 2d; Supplementary Data 3, 4). For instance, a novel isoform of *Sgk3* and a novel gene example upstream of *Tlk2* illustrated with IGV (v2.4.4) view showed that these newly identified transcripts were enriched in apparent H3K4me3 peaks around their TSSs, and these transcripts were also validated by short-read sequencing data (Fig. 2e, f). However, it was worth noting that some high-confidence novel isoforms with previously annotated TSSs were ignored here according to the criteria of H3K4me3 enrichment because the H3K4me3 peaks for these novel transcripts overlapped with the classic TSSs of annotated genes. Besides, for those potential novel genes not accord with our high-confidence criteria, 472 transcripts possessed two or more exons. We also compared the high-confidence TSSs from blastocyst with a published cap analysis gene expression (CAGE) data from mouse ESCs[32], which is similar with in vivo blastocyst stage. Over 80% of potentially novel isoforms with novel TSSs identified from long-read data were supported by CAGE data from ESCs; about 40% of novel genes identified by long-read data from blastocyst were also supported by CAGE data from ESCs (Supplementary Fig. 6e). The non-supported transcripts with novel TSSs by CAGE data might be elicited by the differential expression between ESCs and blastocyst. To further validate these novel transcripts, we randomly selected 10 novel genes and 10 novel isoforms (TPM > 10 in at least one sample, with more than two exons) for PCR amplification and Sanger sequencing. For the novel genes, the exon junctions were all successfully verified (Supplementary Data 5); for the novel isoforms, the exon junctions spanning novel exons and annotated exons were also successfully confirmed (Supplementary Data 6), demonstrating that our newly identified transcripts are highly reliable.

Next, we augmented the GENCODE annotation with the novel transcripts derived from merged transcripts. To assess how different transcriptome annotation can affect the quantification of transcripts, we quantified the mapping reads of the GENCODE-annotated transcriptome and augmented transcriptome using salmon (v0.10.0). We observed that for GENCODE-annotated transcripts possessing potential novel splicing isoforms, about 20% of these transcripts showed large mapping reads changes ($\delta > 5$) when using the augmented annotation as reference ($P < 1e$ −100, Wilcoxon matched-pairs signed rank test; Supplementary Fig. 6f, g). Compared with the expression of the GENCODE-annotated transcripts, the expression of the newly identified transcripts also showed stage-specific expression patterns across preimplantation development (Supplementary Fig. 6h, i), and the expression of these novel transcripts can be integrated to generate a more complete transcriptome landscape. Thus, we identified a large number of novel long-read transcripts, including thousands

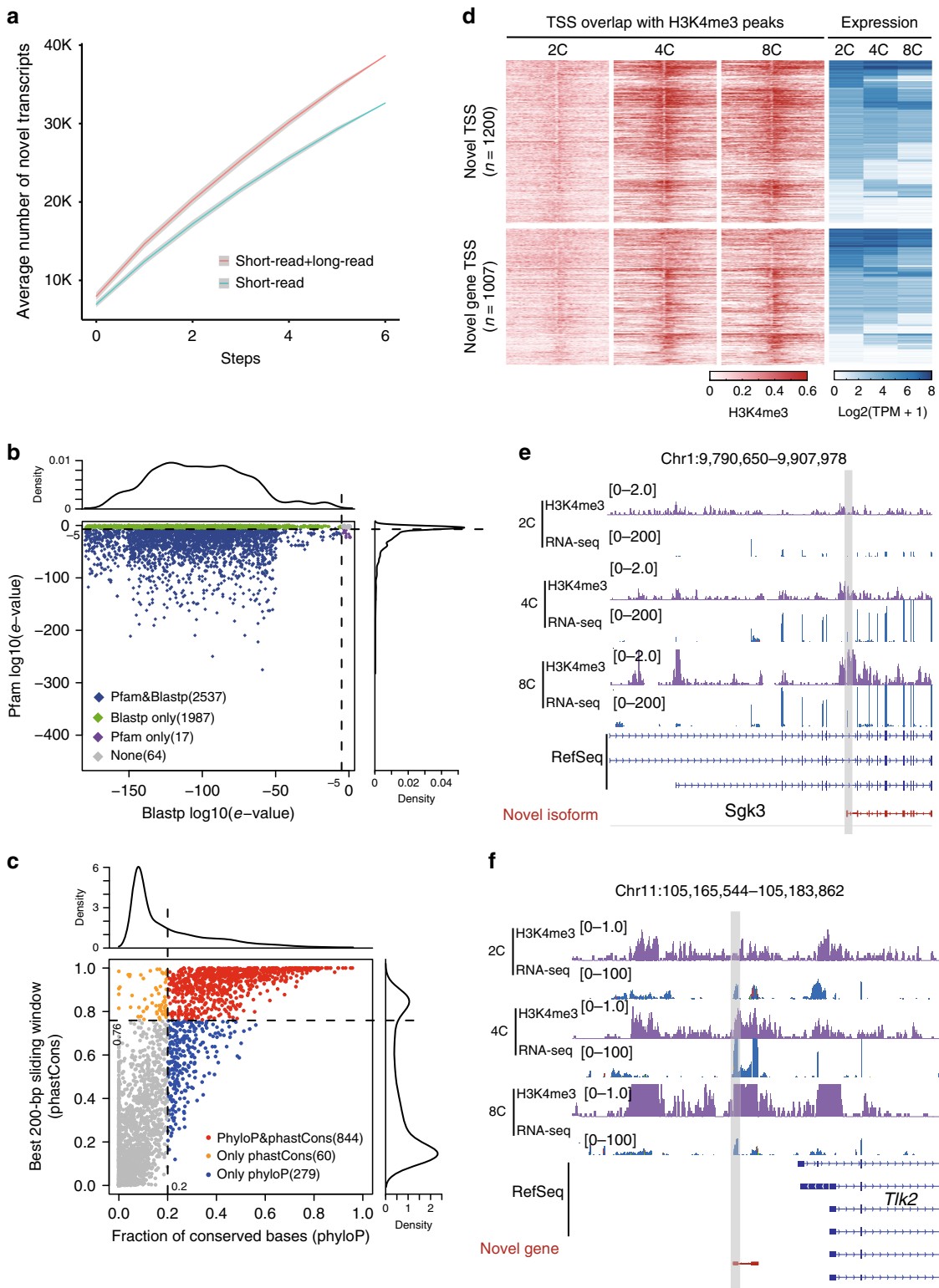

of potential high-confidence novel isoforms and genes, and improved the annotation of preimplantation transcriptome.

**Characterization of alternative splicing in mouse embryos.** Next, we characterized the potentially alternative isoforms in preimplantation embryos with long-read data. We first extracted exon–intron splicing junctions (SJs) from long-read transcripts across the seven stages. Interestingly, more than 97% of SJs showed the canonical GT/AG motifs and less than 2% of SJs

presented noncanonical motifs spanning splicing donor-acceptor sites in all stages (Supplementary Fig. 7a), which was consistent with a previous study[33]. In addition, the majority of SJs derived from long-read transcripts could be validated with short-read sequencing data (Supplementary Fig. 7b), supporting the reliability of the newly identified splicing isoforms.

We then calculated the number of seven types of alternative splicing (AS) events (including skipping exons (SE), alternative 5′ and 3′ splice sites (A5/A3), retained introns (RI), mutually

**Fig. 2 Validation of transcripts identified from long-read data in preimplantation mouse embryos. a** Saturation of novel transcripts identified by short-read sequencing or by a combination of short- and long-read sequencing. This combination was archived by merging transcripts identified from short- and long-read data. The identified novel transcripts were annotated with the GENCODE annotation at each step. The lines and bands represent the mean and the 99% confidence interval of the number of novel transcripts identified at each step, respectively. **b** Sequence homology and domain analysis for novel coding transcripts. The blue prism represents significance ($\log10(e$-value$) < -5$) according to both Blastp and Pfam. Green points indicate Blastp only, purple points indicate Pfam only, and gray points indicate no significance in either analysis. **c** The scatter plot shows the fraction of conserved bases (base-wise phyloP score > 0.972) (x axis) and the maximal 200-bp window average phastCons score (y axis) of novel non-coding transcripts. Blue points indicate transcripts with higher base-wise conservation (phyloP) relative to random control regions. Orange points indicate transcripts with higher window-based conservation (phastCons) relative to random control regions. Red points indicate transcripts that met both conservation criteria. **d** Association between H3K4me3 enrichment and gene expression. Red heatmaps represent the distributions of the H3K4me3 signals in the promoters of novel transcripts with novel TSSs within the annotated loci as well as novel genes overlapped with H3K4me3 peaks (±500-bp). Each row represents a promoter region of ±4 kb around the TSSs for 2-cell, 4-cell, and 8-cell. Blue heatmaps represent the distributions of TPM in the two classes of TSSs. Each row represents the TPM calculated by short-read data form 2-cell, 4-cell, and 8-cell. **e, f** Validation examples of identified transcripts. IGV view of the H3K4me3 density and RNA-seq alignment density in a novel isoform (**e**: Chr1:9790650–9907978; *Sgk3*) and a novel gene (**f**: Chr11:105,165,544–105,183,862).

exclusive exons (MX), and alternative first and last exons (AF/AL)) (Fig. 3a) among the SJ-validated spliced isoforms using SUPPA2 (v2.2.1)[34], and identified approximately 1000 AS events for each stage (Fig. 3b; Supplementary Data 7). In total, 5714 AS events of seven types were identified across the seven stages from the merged transcripts. Moreover, the distribution of AS events recovered from the long-read data was generally consistent with the GENCODE annotation (Supplementary Fig. 7c). We also examined the number of AS genes associated with each type of AS event (described in the Methods section) and found that the proportions of the AS genes of the seven types identified from long-read data ranged from 34.4% ($n = 670$) to 43.2% ($n = 935$) (Fig. 3c). Similarly, the distribution of the AS genes was consistent with that of AS events of the seven types at each stage (Fig. 3b, c) and in the merged condition (Supplementary Fig. 7c, d). In addition, we clustered all samples according to the percent spliced in index (PSI) (Supplementary Fig. 8a, b) and found that the clustering pattern obtained from splicing events was similar to the clustering heatmap generated on the basis of gene expression (Supplementary Fig. 8c, d), indicating that similar to stage-specific expression, stage-linked splicing is associated with embryonic development.

Next, we analyzed the dynamic changes of transcripts in consecutive stages during embryonic development. By comparing the transcriptome in a specific stage with its previous stage, we identified gained transcripts that were absent in a previous stage while present in a current stage, as well as lost transcripts that were present in a previous stage while absent in a current stage. The percentages of gained and lost transcripts were balanced in the majority of consecutive stages; however, 73.4% of gained transcripts and 58.4% of lost transcripts were identified at the 1-cell to 2-cell transition, which is the key stage for zygote genome activation (ZGA) (Fig. 3d). Accordingly, we postulated that the large changes in transcripts might result from global changes in AS events during embryonic development. Consistent with our hypothesis, we also found great changes in AS events of the seven types during embryonic development (Fig. 3e–k).

To explore the potential factors involved in splicing changes, we analyzed the expression of 297 splicing factors[35] in embryonic development. Interestingly, these splicing factors showed stage-specific, but rare, consecutive expression patterns (Fig. 4a; Supplementary Data 8), which was in accordance with the dynamic and stage-specific AS events. For instance, the expression of several splicing factors, such as *Pqdb1*, *Ddx20*, *Sf1*, and *Zqr1*, was remarkably upregulated during 1-cell to 2-cell transition (Fig. 4a), indicating that these splicing factors -linked splicing events might be involved in ZGA process. Moreover, the expression levels of some essential splicing factors, including *Raly* (negative), *Phf5a* (positive), and *Snrpd3* (positive), were

negatively or positively correlated with the number of AS events (Fig. 4b–d).

To trace the dynamic changes of AS across preimplantation development, we first identified the differential alternative splicing (DAS) events of seven types using SUPPA2[34] in consecutive stages from the oocyte to the blastocyst (Supplementary Fig. 9a–g; Supplementary Data 9). Among the identified significant differential alternative splicing events, there were 391 SE events, 224 A5 events, 129 A3 events, 38 RI events, 9 MX events, 580 AF events, and 74 AL events. Next, the significant differential alternative splicing events were traced in consecutive stages from the 1-cell stage to the blastocyst. The river plot showed major global changes in significant differential alternative splicing events (PSI upregulation: $\Delta$PSI $> 0$, $P < 0.05$, which means the fraction of the alternative exon upregulated; PSI down-regulation: $\Delta$PSI $< 0$, $P < 0.05$, which means the fraction of the alternative exon downregulated) during embryonic development (Fig. 4e; Supplementary Data 9, Supplementary Table 1). Intriguingly, rare significant differential alternative splicing events were retained from the 1-cell to blastocyst transitions, regardless of the dynamics from significant to non-significant differential alternative splicing events or vice versa, or between significant upregulated and downregulated differential alternative splicing events (Fig. 4e). Further GO enrichment analysis of the genes associated with these significant differential alternative splicing events across developmental stages demonstrated that these events had important roles in cellular metabolic process, RNA metabolic process, and RNA processing (Fig. 4f; Supplementary Table 2). To identify the potential novel transcripts involved in the ZGA process, the novel genes ($n = 97$) or isoforms ($n = 274$, such as *Ggnbp2*, *Rps23*, *Xiap*, *Ehbp1*) that were upregulated during the 1-cell to 2-cell transition were clustered, showing transient or constitutive activation beyond 2-cell stage (Fig. 4g; Supplementary Data 10). Taken together, these data demonstrate the potential relationship between stage-specific splicing factors and the dynamic changes of alternative splicing events during the preimplantation embryo development.

**Identification of novel transcripts in early embryogenesis.** Among the genes that were upregulated during the 1-cell to 2-cell transition, which were potentially involved in early embryogenesis, we found a novel isoform of *Kdm4dl* and a novel gene designated *XLOC_004958* among the list of the significantly upregulated novel transcripts (Supplementary Data 11). We first PCR-amplified fragments within two neighboring exons using primers spanning two contiguous exons according to our long-read sequencing annotation of *Kdm4dl* and *XLOC_004958* (two splicing isoforms). As shown in Fig. 5a, the exon junctions of both genes were successfully validated by Sanger sequencing.

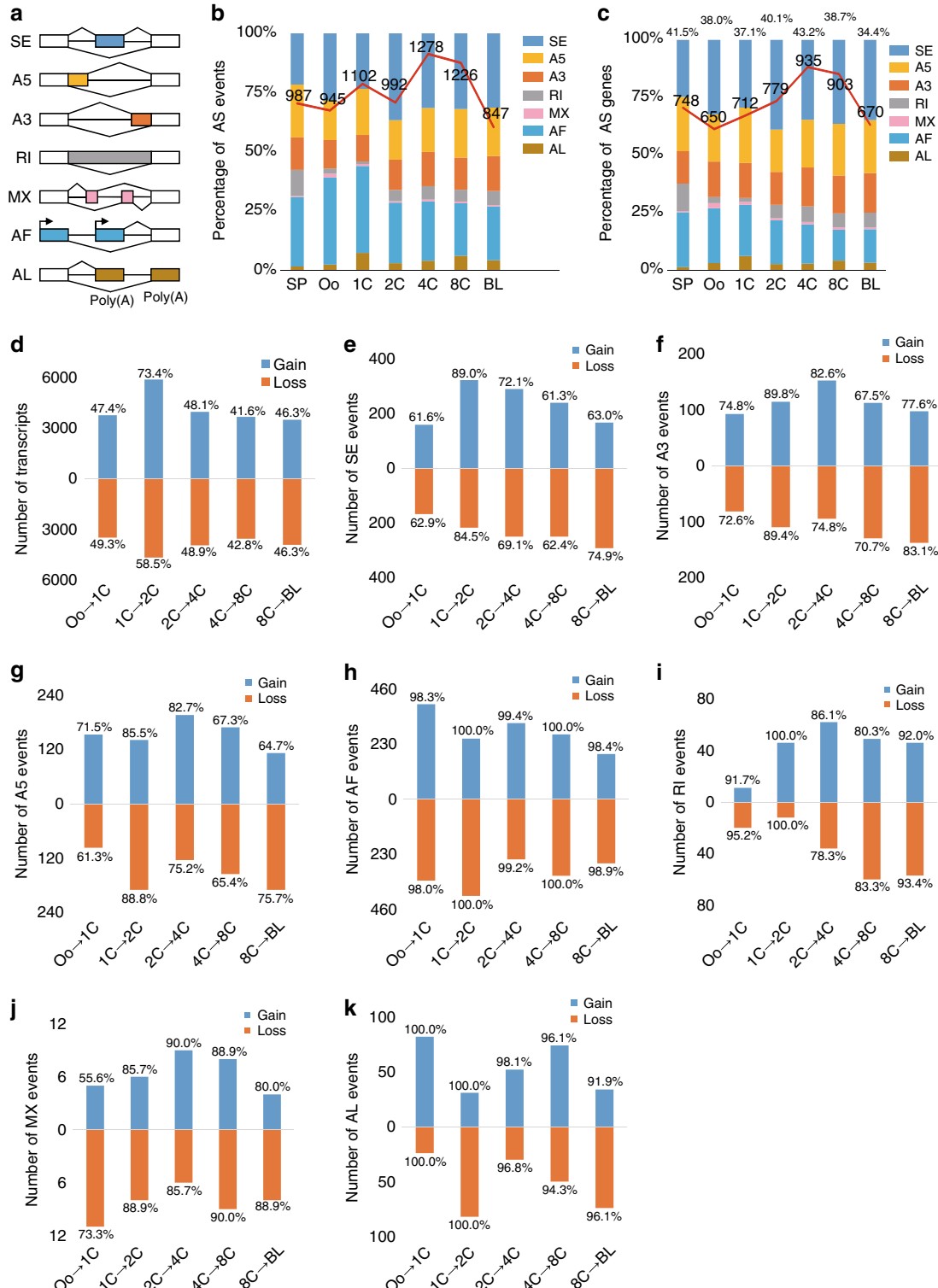

**Fig. 3 Identification of alternative splicing (AS) events and differential splicing events in seven stages of preimplantation mouse embryos. a** Schematic diagram of the seven types of AS events. **b** Distributions of AS events in the seven stages. The percentages (bar) and total numbers (red line and indicated numbers) of AS events in the seven classes are presented in each stage. **c** Distribution of AS genes in the seven stages. The percentages (bar) and total numbers (red line and indicated numbers) of AS genes in the seven classes are presented in each stage. The percentages at the top of the bars represent the proportions of total AS genes among total genes with multiple isoforms in each stage. **d** Numbers of long-read transcripts gained or lost in consecutive stages. The percentages at the top of the bars represent the proportions of gained transcripts among the total transcripts of the later stage, and the percentages at the bottom of the bars represent the proportions of lost transcripts among the total transcripts of the previous stage. **e–k** Numbers of AS events of seven types, which were gained or lost in consecutive stages. The percentages in the top of bars represent the proportion of gained events relative to the total AS events in the later stage, and the percentages in the bottom of bars represent the proportion of lost events relative to the total AS events in the previous stage. **e** for SE, **f** for A3, **g** for A5, **h** for AF, **i** for RI, **j** for MX and **k** for AL.

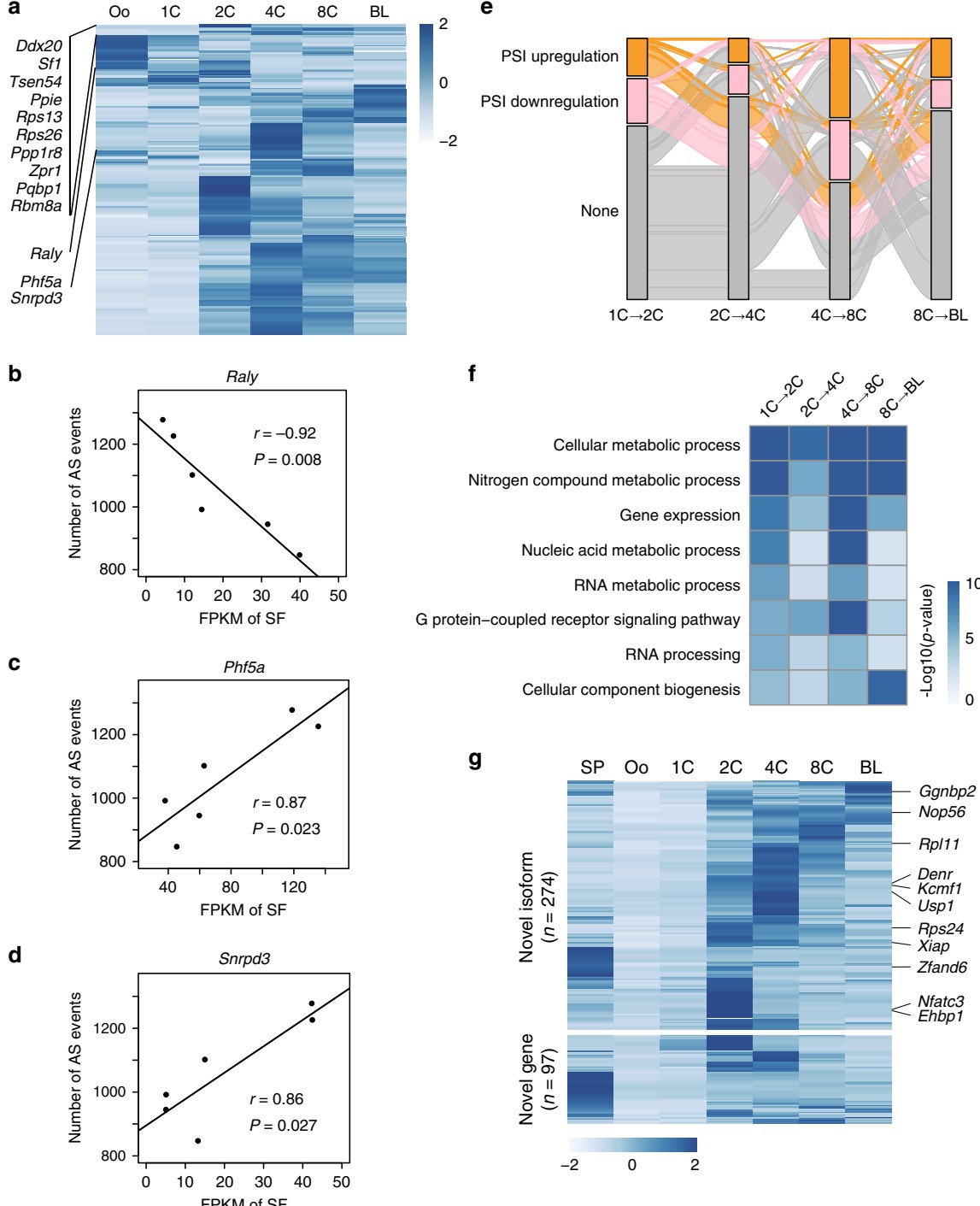

**Fig. 4 AS dynamics during preimplantation embryo development. a** Heatmap showing the expression of splicing factors in six stages. Heatmap shows Z-scores of FPKM by row. The representative splicing factors that were activated during 1-cell to 2-cell transition were also presented. **b–d** Pearson correlation between expression of *Raly* (**b**), *Phf5a* (**c**), and *Snrpd3* (**d**) and number of AS events in six stages (from oocyte to blastocyst). **e** The global dynamics of differential splicing events during early embryo development. Each column represents the tracking trace for differential AS events between adjacent stages. Each bar represents the class of differential splicing events. Up: PSI upregulated; down: PSI downregulated; none: AS events showing no difference. **f** The GO enrichment analysis of biological processes for differential splicing events in adjacent stages. **g** Heatmap showing the expression of novel transcripts (including novel isoforms and novel genes) that was upregulated during the 1-cell to 2-cell transition. The heatmap shows the Z-score of TPM by row.

Intriguingly, both genes began to express from the 2-cell stage and were downregulated at the blastocyst stage (Fig. 5a, b). *XLOC_004958* was predicted to be a novel non-coding gene from chromosome 15 (59100339–59103973); *Kdm4dl* was previously considered a pseudogene, and we redefined the reading frame of

this gene as a novel isoform here (Fig. 5c). According to our annotation, the protein sequence of *Kdm4dl* was highly similar (74.7%) to that of human *KDM4E*, a key histone demethylase (Fig. 5c). To test our hypothesis that *Kdm4dl* and *XLOC_004958* might be involved in preimplantation embryo development, we

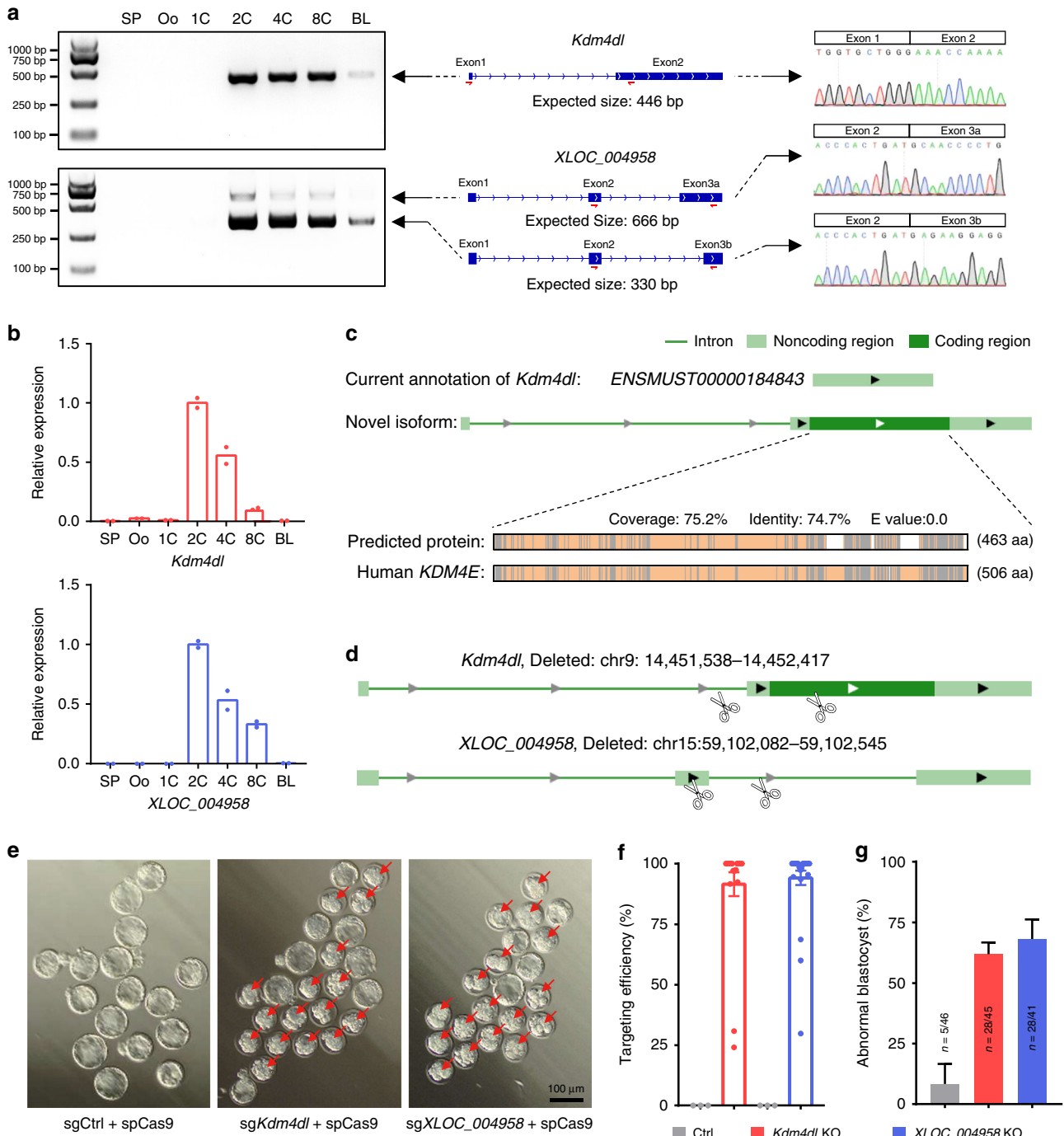

**Fig. 5 Two novel transcripts functionally involved in early embryogenesis. a** RT-PCR validation of *Kdm4dl* and *XLOC_004958* during embryogenesis. The isoform structures of *Kdm4dl* and *XLOC_004958* are shown in the middle; red arrows indicate the loci of the PCR primers, and the sizes of RT-PCR products are displayed. Sanger sequencing chromatograms of the RT-PCR products confirmed the splice junctions in the right panel. Representative data of two independent experiments is shown. **b** The relative expression of *Kdm4dl* and *XLOC_004958* was measured by qPCR analysis. n = 2 biologically independent experiments. **c** Structural schematic showing the comparison of the previously annotated coding frame of *Kdm4dl* and the newly defined coding frame of *Kdm4dl* according to long-read data. In the lower panel, the newly defined protein sequence of *Kdm4dl* was compared with human *KDM4E*. Gray lines represent mismatched amino acids, white lines represent gaps, and light-yellow lines represent overlapping amino acids. **d** Diagram illustrating the CRISPR-mediated knockout of *Kdm4dl* and *XLOC_004958*; the scissors indicate the loci of the guide RNAs (sgRNA sequences are provided in Supplementary Table 3). **e** Morphological imaging of *Kdm4dl* and *XLOC_004958* knockout embryos. The red arrows indicate abnormal blastocysts. The sgRNA-targeting GFP was defined as sgCtrl. Representative data of two independent experiments is shown. **f** The targeting efficiency of *Kdm4dl* and *XLOC_004958* knockout embryos was calculated from Sanger sequencing results by TIDE (https://tide.deskgen.com/) and shown as the mean ± S.E.M. n = 3 embryos for Ctrl, n = 20 embryos for *Kdm4dl*, and n = 28 embryos for *XLOC_004958*. **g** The proportions of abnormal blastocysts following *Kdm4dl* and *XLOC_004958* knockout in E4.5 embryos were calculated. n = 2 biologically independent experiments.

constructed a pair of single-guide RNAs (sgRNA) targeting *Kdm4dl* and *XLOC_004958*, respectively (Fig. 5d). Next, a pair of transcribed sgRNAs and *Streptococcus pyogenes* Cas9 (SpCas9) mRNAs were co-injected into zygotes, followed by in vitro development. Compared with the non-target control group, the efficient knockout of *Kdm4dl* or *XLOC_004958* by using the CRISPR/Cas9 system resulted in delayed or abnormal embryo development (Fig. 5e–f), and the proportion of abnormal blastocysts at E4.5 was significantly increased upon *Kdm4dl* or *XLOC_004958* depletion (Fig. 5g). Taken together, we identified two novel transcripts that were functionally involved in pre-implantation embryogenesis.

## Discussion

Long-read sequencing technology has been widely used for annotating novel transcripts and splicing events in modeling organisms and cancer tissues with numerous cell numbers[16–19]. In the present study, we developed a computational pipeline that integrates single-cell RNA-seq-based SMART-seq2[36] and cDNA-based PacBio long-read sequencing, making great process in the resolution and accuracy of transcriptome annotation during mouse preimplantation embryo development. We aimed to elucidating potential novel transcripts and strain-specific splicing isoforms as well as events involved in early embryogenesis. In summary, we identified thousands of novel presumptive protein-coding transcripts and thousands of novel isoforms of GENECODE-annotated genes (Fig. 1). Although long-read sequencing has been used to profile the ZGA of zebrafish embryo[37], our findings greatly expanded the transcriptome annotation for the mammalian community, with a more diverse transcriptional and regulatory repertoire.

We identified 2280 potential transcripts from previously unannotated loci and 6289 potentially novel isoforms from annotated gene loci. In addition to validation by using short-read sequencing data, Sanger sequencing, and functional domain conservation analysis, we also verified the newly identified transcripts according to the significant enrichment of active H3K4me3 modifications, which are associated with transcription initiation[38], around TSS regions (Fig. 2). With this strategy, we stringently identified 1200 high-confidence novel isoforms with new TSSs and 1007 high-confidence novel genes (Fig. 2d). Even though, there are some limitations remained because of the lack of perfect methods for definitely validating these potential novel genes, and the functional and biological significance of these novel transcripts (except the previously undescribed isoform of *Kdm4dl* and the novel gene *XLOC_004958* addressed in the present study) needs further exploration in our future study. Taking advantage of long-read sequencing, we also profiled the full RNA splicing landscape of the preimplantation embryo and recovered thousands of novel alterative splicing junctions, with mainly of SE, A5, and AF splicing types (Fig. 3), supported by perfect canonical GT/AG splicing motifs (Supplementary Fig. 7a). By integrating newly identified transcripts with short-read sequencing data, we generated an augmented transcriptome with more complete and precise contents for preimplantation embryo (Supplementary Fig. 6f–g; Supplementary Data 12). Importantly, we observed a high percentage (over 40%) of transcripts with a gain or loss of alternative splicing between the two consecutive stage transitions (Fig. 3d), which showed highly dynamic and discontinuous characteristics across these stage transitions, highlighting the important roles of alternative splicing factors, genes, and events (Fig. 3). We also proposed a series of previously undescribed alternative splicing isoforms or genes inferred from the long-read data that were specifically activated during ZGA (Fig. 4g), thus expanding the candidate factors involved in ZGA progression. We also validated two novel transcripts: an undefined isoform of *Kdm4dl* and the gene *XLOC_004958*, which are functionally involved in early embryogenesis (Fig. 5). However, the mechanisms underlying their functions, possibly involving histone demethylation or potential targets of the non-coding gene *XLOC_004958*, require further investigation in the future.

However, the low-cell number-based long-read sequencing strategy has limitations for the profiling of short and non-poly(A) transcripts[39], and many unusual transcripts, such as repeats and translocations, were excluded during the analysis, therefore, optimized sequencing technology and analysis methods are desired for the constructing of a more complete transcriptome. In addition, we subjected enough cDNAs to long-read sequencing to avoid size selection and bias generation, whereas newer Iso-Seq technology may help to produce higher-quality data with a reduced DNA input.

In summary, we applied our analysis pipeline using both long- and short-read sequencing on hybrid crosses to identify thousands of high-confidence potential novel genes and isoforms in pre-implantation embryos, providing a valuable resource for clarifying the molecular basis of early embryogenesis. Thus, our findings expand the transcriptome and splicing annotation available for the mammalian community and could help to explore the mechanisms underlying abnormal gamete development.

## Methods

**Mouse sperm, oocytes, and embryo collection.** All experiments involving mice were approved by the Animal Care and Use Committee of the Institute of Neuroscience, Chinese Academy of Sciences, Shanghai, China. Mice were maintained in an Assessment and Accreditation of Laboratory Animal Care credited specific pathogen-free facility under a 12 h light, 12 h dark cycle. Ambient temperature is 20 °C, relative humidity is 50%. Sperm were collected from the epididymis of DBA/2 males. MII oocytes were derived from 3- to 5-week-old C57BL/6J females induced to superovulate via the injection of 7.5 IU pregnant mare serum gonadotropin (PMSG, San Sheng), followed by the injection of 7.5 IU human chorionic gonadotropin (hCG, San Sheng) 48 h later. Embryos were collected from the oviduct of the superovulated females after mating them with DBA/2 males. The oocytes and embryos were isolated in M2 medium (Sigma) at 20 h post hCG injection. Other embryos were cultured in KSOM media (Merck Millipore) at 37 °C with 5% $CO_2$ and collected at the following time points: zygotes (24–26 h post hCG), two-cell stage (46–48 h post hCG), four-cell stage (54–58 h post hCG), eight-cell stage (68–70 h post hCG), and blastocyst stage (94–96 h post hCG).

**In vitro transcription.** The SpCas9 expression vector was linearized with the *Bbs*I enzyme (NEB) and transcribed in vitro using a T7 Ultra Kit (Ambion) according to the manufacturer's protocols. mRNA was purified by using a Mini Kit (Qiagen). sgRNA oligos were annealed to pUC57-sgRNA expression vectors with the T7 promoter. Then, sgRNAs were amplified and transcribed in vitro by using the MEGAshortscript Kit (Ambion). The sgRNAs were purified with the MEGAclear Kit (Ambion) according to the manufacturer's protocols.

**Microinjection of mouse zygotes.** Female mice at 4 weeks of age were super-ovulated and mated with male mice. Fertilized one-cell embryos were collected from the oviducts. For microinjection, mRNA mixtures containing two sgRNAs (50 ng/µl) and Spcas9 (100 ng/µl) were injected into the cytoplasm of zygotes in a droplet of M2 medium containing 5 µg/ml cytochalasin B (CB) using a piezo (Primetech) microinjector. The injected zygotes were cultured in KSOM media at 37 °C under 5% $CO_2$ in air until E3.5 for genotyping.

**Embryo lysis and genotyping.** Each individual embryo was transferred to one well of an eight-well PCR strip and lysed using the Mouse Direct PCR Kit (Bimake) according to the manufacturer's protocols. The target region was PCR amplified with Phanta® Max Super-Fidelity DNA Polymerase (Vazyme).

**cDNA preparation.** Embryos were lysed in 50 µl of guanidine isothiocyanate solution (Invitrogen, 15577-018) at 42 °C for 15 min. Nuclease-free water was added to 200 µl, and 3 volumes of absolute ethanol, a 1/10 volume of 3 M acetate sodium (Invitrogen, AM9740) and 2 µl of glycogen (Roche, 10901393001) were added, followed by uniform mixing. After storage at −80 °C for 2 h, the total RNA was pelleted by centrifugation at 12,000×*g* for 40 min. The total RNA pellets were

dissolved in lysis solution and used as a template for the first step of cDNA synthesis, and cDNA was amplified via the Smart-seq2 protocol[22,36].

**PacBio SMRT library construction and sequencing**. Two micrograms of each amplified cDNA was transformed into SMRTbell templates using the PacBio SMRTbell Express Template Prep Kit. The libraries were sequenced using the PacBio Sequel System at the Berry Genomics Company (Beijing, China).

**RNA-seq library preparation and sequencing**. The sequencing libraries were constructed using the TruePrep DNA Library Prep Kit V2 for Illumina (Vazyme Biotech, TD502-01) according to the manufacturer's instructions. All libraries were sequenced on the Illumina NovaSeq 6000 platform in 150-bp paired-end mode at the Berry Genomics Company (Beijing, China).

**RNA-seq data pre-processing and quality control**. The quality control of illumina RNA-seq data was done by FastQC (v0.11.8). The RNA-seq data were trimmed by TrimGalore (v0.6.1) with default parameters for pair-end data. The trimmed RNA-seq data were aligned to the mm10 reference using STAR (v2.5.0a)[40] with parameters --twopassMode Basic --outSAMtype BAM Unsorted --out-SAMstrandField intronMotif.

The short-read transcripts were assembled using Cufflinks (v2.2.1)[41] with default parameters in each stage, only transcripts with FPKM > 1 were retained.

**Long-read sequencing data pre-processing**. PacBio Iso-seq3 pipeline was used for generating full-length (FL) transcripts with default parameters. First, we generated circular consensus sequences (CCS) from subread sequences, and clipped their 5′ and 3′ primers and polyA tails. Then, we clustered CCS reads and generated unpolished transcripts. Third, we polished transcripts using subreads, and only retained the high-quality transcript sequences for subsequent analysis.

The high-quality transcript sequences were mapped to mouse genome assembly mm10 by GMAP (version 2019-03-04)[42] with parameters −n 0 −z sense_force. Then the cDNA_Cupcake package (https://github.com/Magdoll/cDNA_Cupcake, v6.6) was used for collapsing redundant isoforms, filtering collapse results by minimum FL count support, and filtering away 5′ degraded isoforms with default parameters.

**Annotation of long-read transcripts**. We used the Cuffmerge (v2.2.1)[41] to merge all the long-read transcripts identified from sperm to blastocyst. Long-read transcripts were annotated by comparing with mouse GENCODE annotation (vM20) using Cuffcompare (v2.2.1)[41], and classified the long-read transcripts into five classes according to their most closely matching GENCODE transcript. The Cuffcompare class codes used in the categorization of the long-read transcripts were defined as follows: "=" for "Exact match to annotation", "c" for "Sequential subset of exons contained within annotation", the set of "e" and "j" for "Potentially novel isoform", "u" for "Potentially novel gene" and the other codes for "Other".

The set of "=" and "c" was defined as "annotated transcripts", and the set of "e", "j", and "u" was considered as "novel transcripts". For annotated transcripts, we divided them into "protein coding", "long non-coding RNA (lncRNA)" and "else" according to GENCODE annotation. For novel transcripts, we estimated their protein-coding potential using CPAT (v2.2.0) with the default cutoff (cutoff = 0.44) for mouse[23]. The novel transcripts with high protein-coding potential (cutoff >0.44) were defined as "protein coding". The novel transcripts with low protein-coding potential (cutoff ≤ 0.44) and transcript length >200-bp were defined as "lncRNA". The others were defined as "else".

To analyze the transcript dynamics during preimplantation development, we used Cuffcompare to compare long-read transcripts from adjacent stages. We defined transcripts with class code "=" or "c" in adjacent stages as "shared" transcripts, the transcripts with other class codes and in the preceding stage were defined as "Loss" transcripts, and the transcripts with other class codes and in the latter stage were defined as "Gain" transcripts.

**Quantification of long-read transcripts**. We estimated the expression of long-read transcripts and genes with TPM (Transcripts Per Million) and FPKM (fragments per kilobase million), respectively. Salmon (v0.10.0)[43] was used to calculate TPM with parameter −l A for RNA-seq pair-end data. StringTie (v1.3.3b)[44] was used to calculate FPKM with default parameter for RNA-seq pair-end data.

The transcript or gene that did not express in oocyte (TPM ≤ 1 or FPKM ≤ 1), but expressed in 1-cell and 2-cell, and the expression of 2-cell was more than twice of 1-cell, was defined as ZGA transcript/gene.

**Analysis of potential novel transcripts**. We defined the "protein coding" transcripts from novel transcripts as novel coding transcripts. The "lncRNA" and "else" from novel transcripts were considered as novel non-coding transcripts. For novel coding transcripts, we used TRANSDECODER (v5.5.0)[45] to extract ORFs from these transcripts, and limited the length of ORF to be at least 300-bp, 4605 transcripts were identified to have at least one ORF. The obtained ORFs were mapped to UniProt (release 2019_08)[25] and PFAM-A (v31.0)[28] using BLASTP (v2.9.0)[26], and hmmer (v3.2.1)[27], respectively. The resulted Pfam domains were further filtered with the procedure described in a recent study[37]. In addition, we performed

GO enrichment analysis in biological processes using STRING (v11.0)[46] for the set of known homology proteins corresponding to the novel coding transcripts. The homology proteins were defined as known proteins corresponding to the ORFs of novel coding transcripts with ident=100 in Uniprot. For novel non-coding transcripts, we calculated phyloP[29] and phastCons[30] scores using bigWigAverageOverBed (v2) (from UCSC) with phyloP and phastCons data of 60 vertebrate species from UCSC, respectively. We also measured the conservation of random control regions using a length-matched set of inter-genic sequences. We compared the difference of median in average conservative scores between novel non-coding transcripts and sequences of random control regions with two-tailed Wilcoxon rank sum test. Similar to the procedure described previously[37], we measured the fraction of significantly conserved bases (phyloP, score > 0.972) and the maximally conserved 200-nt sliding window (phastCons), the phyloP score was chosen as the 95th percentile of all scores in the randomly control regions. And for both base-wise conservation (phyloP) and contiguous window conservation (phastCons) analyses, cutoffs for significant transcripts were chosen as the 95th percentile of all scores of sequences in random control regions (0.2 for phyloP and 0.76 for phastCons).

**Comparison between long-read data and short-read data**. To assess the effect of novel transcripts identified by long-read sequencing, we compared the saturation of novel transcripts identified by short-read sequencing or by combination of short-read and long-read sequencing. We first randomly selected short-read data from a stage, and identified novel transcripts using Cuffcompare with GENCODE annotation. Second, short-read transcripts from a single stage were merged with previously identified transcripts using Cuffmerge, and then annotated with GENCODE annotation using Cuffcompare to identify novel transcripts. The computation was complete when all of the short-read data from seven stages had been included in the analysis. The above procedure was repeated 100 times at random, the mean and 99% confidence interval of the number of novel transcripts were calculated at each point when new short-read data was introduced. The saturation analysis was also done for combination of short-read and long-read sequencing as the same. The only difference was that we first merged long-read transcripts and short-read transcripts at each stage.

To assess the quantification changes of transcriptome caused by novel transcripts identified by long-read sequencing, we used salmon to quantify the mapped reads of transcriptome using only GENCODE annotation or GENCODE annotation augmented with long-read transcripts as reference. For this two transcriptome references, we compared the number of mapped reads, and only the transcripts whose numbers were greatly changed ($\delta > 5$) at least one stage were shown.

Average percentage of bases and motifs in exons were calculated in our custom python scripts. We used a perl script obtained from http://caballero.github.io/SeqComplex/ to calculate various characterization of sequence complexity, including Markov model features and sequence composition. The exons were overlapped with low complexity region obtained from RepeatMasker in UCSC Table browser to assess the complexity.

**Transcript validation**. We categorized the transcriptional start sites (TSSs) into three groups based on transcript classification. The ±300-bp around the TSS in GENCODE annotation was defined as the annotated TSS interval. TSSs of "Potentially novel gene" transcripts were defined as novel gene TSSs, TSSs of "Potentially novel isoform" transcript and overlap with the annotated TSS interval were defined as classic TSSs, the remaining TSSs were defined as novel TSSs. For comparing, we defined the TSS of "Exact match to annotation" transcript as annotated TSSs.

We obtained histone modification H3K4me3 data of 2-cell, 4-cell, and 8-cell from a previous study[5], and defined ±4 kb around the TSS as the promoter region. We divided promoter regions into 161 bins by using 100-bp as the length of each bin and 50-bp as the length of step, and calculated the normalize H3K4me3 score of each bin using bigWigAverageOverbed. Peaks of H3K4me3 were called using MACS2 (v2.0.10.20131216)[47] with the call peak function using parameters -g mm −B −q 0.05 --nomodel --broad --shiftsize=73 −SPMR. We merged all the H3K4me3 peaks of 2-cell, 4-cell, and 8-cell, TSS that overlap with peaks in its ±500-bp consider as high-confidence TSS.

We obtained CAGE bigwig data of ESC from a previous study[32], and extracted the interval where TPM was not 0 in any samples, TSS that overlap with these interval in its ±500 bp was considered verified by CAGE data. We compared the TSS of blastocyst with CAGE data.

**Comparison of transcripts in oocyte**. We merged the transcripts identified from short-read data or long-read data of oocyte using Cuffmerge, specially, the transcripts without strand that identified from short-read data were filtered. And compared them with oocyte annotation in a previous study[24]. Our data identified fewer numbers of transcripts (15,271 transcripts vs. 82,939 transcripts and 10,190 genes vs. 39,099 genes) in oocytes. Transcripts with class code "=" and "c" annotated by Cuffcompare were treated as shared transcripts between two transcriptomes. 7251 out of 15,271 (47.4%) transcripts were identified by both the two data.

**Identification and characterization of AS DAS events.** Seven types of alternative splicing (AS) events were identified using SUPPA2 (v2.2.1)[34] in each stage, including skipping exons (SE), alternative 5′ or 3′ splice sites (A5/A3), retained introns (RI), mutually exclusive exons (MX), and alternative first or last exons (AF, AL). AS genes were defined as genes associated with AS events. To obtain "high-confidence" splicing junctions, we also identified splicing junctions in short-read data using STAR. Only splicing junctions in long-read transcripts that supported by at least ten short reads in at least one sample were considered as "high-confidence".

The list of splicing factor genes presented in Fig. 4a–d was obtained from a previous study[35]. We calculated the difference of expression of SFs (Padj) between 1-cell and 2-cell using DEseq2(v1.20.0)[48] with raw reads as input. We performed correlation analyses between the expression of splicing factors and the number of AS events via Pearson correlation.

In each stage, we applied SUPPA2 to calculate the PSI of each AS event using RNA-seq data with the default parameters, and merged long-read transcripts were used as a reference. We also applied SUPPA2 to identify differential alternative splicing events in consecutive stages from the oocyte to the blastocyst (excluding the sperm stage). The GO enrichment analysis of genes associated with differential splicing events was performed by using PANTHER (v14.1)[49].

**PCR validation of alternative splicing.** Alternatively spliced genes were PCR amplified using HotStart Taq DNA Polymerase (TaKaRa). The PCR-amplified fragments were separated by agarose gel electrophoresis. The primer sequences used are listed in Supplementary Table 4.

**Other bioinformatic analysis and statistical analysis.** The statistical analysis was performed using a two-tailed Wilcoxon rank sum test and Fisher's exact test. Wald significance tests were applied in DESeq2 differential analysis. All the software and data processing protocol used are listed in Supplementary Note 1.

**Reporting summary.** Further information on research design is available in the Nature Research Reporting Summary linked to this article.

## Data availability

The authors declare that all data supporting the findings of this study are available within the article and its supplementary information files or from the corresponding author upon reasonable request. The long-read and short-read raw sequencing data have been deposited at SRP database under the accession code: SRP225196. The processed data including transcriptome annotation and expression level matrix have been deposited at the GEO database under accession code: GSE138760. CAGE data are obtained from 'GSM3188176', 'GSM3188177', 'GSM3317700', 'GSM3317701'. H3k4me3 data are obtained from 'GSE73952'. Custom oocyte transcriptome annotation is obtained from 'GSE70116'. The source data underlying Figs. 1b–f, 2a–d, 3b–k, 4a–g and 5b, f, g; and Supplementary Figs. 1–9 are provided as a Source Data file.

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

## Acknowledgements

This work was supported by the National Key Research and Development Program of China (2016YFA0500903; 2018YFC1004700), National Natural Science Foundation of China (81830004), the National Natural Science Foundation of China (No. 61873276), the Major Research Plan of the National Key R&D Program of China (No. 2016YFC0901600), and Excellent Youth Foundation of Guangdong Scientific Committee, 2020B1515020018, Y.Q. We thank Dr. Qiang Sun for technical and analytical support.

## Author contributions

Y.Q., S.H., and X.L. designed and performed the experiments; W. S., Y.Q., C. R., S.H., and J.Y. co-wrote the manuscript; C.R. and J.Y. performed the computational analysis; J.F., J.L., S.W., Q.C., and X.B. performed and helped with the experiments and data analysis. X.H., Y.Q., L.Z., and W.S. conceived and supervised this work.

## Competing interests

The authors declare no competing interests.
