## [Peer Review File · Nature Communications]

Reviewers' Comments:

Reviewer #1:

Remarks to the Author:

In their manuscript, "High resolution and allelic-specific annotation of preimplantation embryo transcriptome using long-read sequencing", the anonymous authors carry out transcriptome sequencing of the beginning part of the germinal stage in embryogenesis using PacBio SMRT sequencing. The long-reads SMRT sequencing can produce were leveraged to more fully characterize the mouse preimplantation transcriptome, uncovering many novel putative coding and non-coding transcripts as well as novel putative splicing isoforms. In addition, the authors followed their earlier work (ref 11) on elucidating parent of origin effects to identify allele-specific alternative splicing events where changes in splicing patterns over time as well as levels of expression were demonstrated in an allele-specific way, with suggestions relating an evolving understanding of the zygotic gene activation events that occur as part of early embryogenesis.

Casting a more comprehensive net by employing multiple technologies as was done in this manuscript is critical to advance to a more accurate annotation of a transcriptome, and the authors provide ample evidence that there may exist entirely new aspects of biology that have heretofore gone unobserved. The uncovering of several thousand novel transcripts and novel splice forms in the earliest stages embryogenesis (with the authors reporting > 20% on average of the transcripts detected were novel, a striking result if these all turn out to be validated), along with the demonstration of strong parent-of-origin effects observed in alternative splicing events as well as the dynamics of those events, well demonstrates the potential value of these findings as a general resource that will be of high interest to anybody who thinks about transcriptome data. The question really is how much of the discoveries reported are valid functionally, given no experimental support is given regarding any of the findings playing a functional role, and then given the claims that the long-read sequencing really enabled many of the findings, what role the long-reads played in the general findings reported. For validation, the authors employed what could be a pretty comprehensive set of

bioinformatic analyses to support that the novel transcripts and isoforms were likely functional (comprehensive characterization of homology to existing proteins and domains, evolutionary conservation, epigenetic enrichments in the TSS, and so on). But often the details regarding the results are not clearly expressed and confusing at times, and then there is not very strong statistical rigor throughout, so difficult to interpret the true scope of the findings.

In the specific/general comments below, I have attempted to make clear the above type concerns in some detail.

Specific Comments:

1. Regarding the experimental design (as depicted in fig 1a), to help those like myself less familiar with embryogenesis, it would be nice to indicate how far removed the blastocyst is from the earlier stages that were profiled. It seems like a big jump (from 8 cells to blastocyst) and so understanding the rationale for that would be of interest.

Further, it is not clear even how many embryos were characterized at each stage. In the results the authors indicate 150-200 cells were used for each sample, but it is unclear what constitutes a sample. Does this mean that 150-200 zygotes were sequenced? And then 75 2-cell blastomere (excuse my terminology if not completely correct) and so on? Or were the cells batched in order to get enough cells and then moved forward with the prep step? Clearly if batching was done there is the potential to introduce some noise and a loss of resolution. In any case, this should all be clarified.

2. The dataset appears to have been generated on slightly older technology, given the authors indicated they did a 4-part split in their amplification, which is no longer required for the IsoSeq assay, and in fact carrying out size-selection prior to library construction could create biases in the transcriptional space due to the physical separation carried out prior to sequencing. I'm not advocating in any way that the experiment should be repeated on the

newer tech, but rather think it important to discuss whether this could have created a bias and what the implications of that would be for the results reported, if any.

3. There is some characterization of the novel transcripts and isoforms provided (such as figure 1 and some supp figures), but what seems noticeably absent are summaries relating to the number of exons in the novel transcripts, the exon lengths, the sequence complexity of the exons, and how those statistics compare for the novel transcripts detectable with only short read data versus those requiring long-read data, and then how these compare to GENCODE transcripts. What are the characteristics of those transcripts that were only identified given the long-read data and why couldn't they be assembled from short-read data? Given figure 2a clearly shows the long-reads are important, why not press to explain exactly why?

This seems to get lost in conveying of other results such as 2e, where a novel gene is shown upstream of Tik2, but it looks like this novel transcript could have been easily detected with short reads? So most of the novelty this way is driven by looking in conditions that have not been explored deeply before?

4. I like how the authors quantified the isoforms by mapping short reads to the merged isoform dataset across all stages. However, it appears only HQ transcripts were used, but perhaps it would be better to merge the PacBio subreads and run IsoSeq on the merged raw data? This would likely increase the representation of isoforms/rescue isoforms that likely dropped out in the cluster/refine steps when processed separately.

5. The analysis tools that were used (Cuffcompare, SUPPA, StringTie, CPAT, ect.) all seem fine as I think they had been used in a paper I had always considered as a gold standard for novel transcript annotation:

However, today there are what could be considered as improved packages were released:

<https://www.ncbi.nlm.nih.gov/pmc/articles/PMC4417758/>

and now streamlined into one tool, SQANTI2. I'm sure the authors are familiar with this, and I again am not advocating that they redo everything with these improved tools, but would be interested in whether the authors think these newer tools may provide a more comprehensive and accurate view.

6. Figure 2d is difficult to interpret. The legend or methods/supp do not really explain exactly what was done, what the enrichments represent, how to assess what is statistically significant and so on. For example, in the "novel gene TSS" result the H3K4me3 enrichment pattern does not seem to be well correlated with expression. Is that expected? Further, the correlation for the classic TSS category doesn't seem as strong as for the novel TSS, which doesn't seem right. What about for annotated TSS's? Was this supposed to be represented in supp figure 4? Basically, this should all be clarified more quantitatively with statistical significances as well, conveying what actually is expected by chance versus what is expected as the gold standard.

7. For the GO enrichments such as that depicted in Supp figure 3a, it is unclear how the enrichments were computed, how novel genes were assigned to GO categories. This doesn't appear to be described anywhere in the materials provided. Do the novel genes take on categories that relate to their closest homolog? Or gene with closest sequence identity? Or???

8. In several locations the authors use adjectives such as giant, remarkable, and drastic to describe results, but these are never associated with any statistical tests that would help the reader understand what hypotheses are being tested and why the result reported should be deserved of such adjectives. For example, line 244 in the main text indicates, "...novel isoforms showed giant mapping read change...", with a delta statistic given, but no description on why the statistic given is giant, what would be expected by chance and how significant the observed result actually is.

Another example is line 302, where the authors indicate that the expression of several splicing factors were “remarkably upregulated during 1-cell to 2-cell transition” with reference to supp figure 7h, but this is not really well shown in the figure, given the color scale given for the z-score used to depicted the expression ranged for -2 to 2, so that the max color under the assumption of the values being normally distributed would be only nominally significant and not significant if one were to correct for multiple testing. But no description of statistical significance, false discovery rates or other such characterizations are provided, making it difficult to understand if the results presented are truly “remarkable”.

The same goes for line 315 in which the authors state, “River plot showed drastic global changes of significant DAS events during embryonic development.” Where in fact it is difficult to understand what exactly the river plot (figure 3e) shows in this instance. There are no statistical significances given for the up or down regulation events shown, no significance for the changes in these regulation patterns over the different stages, and then nothing indicated regarding any kind of correction needed for the expansive multiple testing that is happening in these data. None of this is well described in any of the methods material.

9. Further regarding statistical significance, in results such as that depicted in figure 3f, corrected p-values should be shown or false discovery rates so that the reader can understand what was actually expected by chance. Further, for enrichment statistics such as those provided, the “effect sizes” should be given as well (in this case would be a fold-enrichment or odds ratio) since the p values will be correlated with sample size and the very large and broad categories will be detected with higher power and will be less relevant since they really don’t tie you into any specific biology.

10. In lines 327-333 the authors nicely give some PCR validation of a single novel and two novel isoforms. However, these appear to have been selected to highlight out of thousands of novel genes/isoforms detected, but no justification is given as to why these 3 examples were picked for validation. How many validations were attempted? Was there any prioritization given to the examples that were selected?

More General Comments:

1. I have not made any comments relating to grammar to include coherent sentence structure, but significant attention should be paid to significantly smoothing out the text throughout. As an example, in the first sentence the reader encounters (the first sentence of the summary paragraph), the sentence reads, "The development of third generation of long-read sequencing...", which should be corrected to something like, "The development of third generation, long-read sequencing technologies..." or something like that. Grammatical errors, spelling mistakes, and confusing sentence structures occur throughout.

2. For the supplementary material provided as excel spreadsheets, there is no legend or description given, making it nearly impossible to understand what is actually provided, what the different columns represent, how the values were computed and so on. It's clearly a lot of material that would be very useful to have described. Perhaps an issue in the conversion for submission (I got the excel spreadsheets, the supp tables, from the merged zip file that was provided)?

3. Related to the above lack of description around the supplementary material, I did not see anywhere described where the data used to derive all of the results reside. As it stands it is not possible to reproduce any of the results presented without access to those data.

Reviewer #2:

Remarks to the Author:

In this manuscript, the authors re-visit the current genome/transcriptome annotation specifically in the context of pre-implantation mouse development during the window of zygotic genome activation. They perform long-read sequencing by PacBio, compare this with more conventional short read sequencing data, and find a large set of regions that they annotate into defined groups, notably novel transcript isoforms and entirely novel genes.

For the major part, the work in itself is very valuable; however, the manuscript is at times almost incomprehensible and requires major revision for language, syntax and context.

Far fewer numbers should be used in the text. It should be clear how many new transcript isoforms and novel genes were detected, but the constant comparison and percentages of various details is confusing and misleading. This is confounded by the fact that non of the numbers seem to add up. For example, in Figure 1b,c, the proportion of orange and light blue bars does not correspond even in the slightest, and what does "Sequential subset of exons contained within annotation" mean? The description and depiction in Fig. 1a is very clear and understandable, and hence it would make sense to stick to those same 3 groups ("known isoforms", "novel isoforms", "novel genes") throughout. If "other" refers to sequencing errors or other artificial reads that have no biological relevance, they should be eliminated during the first steps of the analysis and not displayed.

Within these many comparisons, the authors conclude that almost half of their novel genes are long non-coding RNAs. Does this relate only to the "novel gene" group, or to all novel isoforms&genes? The fraction seems too high.

The oocyte transcriptome has been custom-annotated before (Gahurova et al., *Epigenetics & Chromatin* 2017). The authors must compare their data to this previous annotation update, in particular with regards to the often unusual exon usage in oocytes.

It is essential that solid verifications are performed to confirm the existence of novel candidate genes/isoforms, I suggest in the order of magnitude of at least 10 each. The very few examples shown in Fig. 3h are not sufficient evidence by any stretch. The bands shown are unconvincing (e.g., Ypel5) and will need to be confirmed by Sanger sequencing to indeed reflect the candidate transcript. Similarly for the XLOC example, the 750 bp band would normally be considered as unspecific and will need sequence-confirmation. If these are the best examples of confirmatory PCRs, they casts significant doubt on the data reported. If indeed the sequencing data suggest the concomitant presence of multiple transcript isoforms

in the same cell/stage in different proportions, then the authors should indicate estimated proportions (from the sequencing data) and still confirm the obtained PCR bands independently by Sanger sequencing.

Figures 4 and 5 are very difficult to follow and should be reworked for being more intuitively comprehensible. Similar to the above comment, splicing isoforms need to be robustly confirmed in individual RT-PCR and Sanger sequencing approaches.

The parent-of-origin section is not meaningful because the data only originate from one particular cross, C57BL/6 mother x DBA/2 father. Thus, beyond the sperm and oocyte stage and in particular with ZGA, strain-specific differences will have a major effect on allelic biases. To make this section of value, the reciprocal cross would need to be analysed as well. If the reciprocal cross is not performed, this part of the analysis should be removed or significantly shortened, and the caveat of strain- over parent-of-origin-induced biases discussed.

Reviewer #3:

Remarks to the Author:

stages of pre-implantation embryo development. The subject seems to be of general interest and the data is likely to be a useful resource. However, there are some problems with the analysis and interpretation of data.

The manuscript has reasonable structure but needs extensive editing for proper English, and improved understandability

My main concerns are:

1. The final set of novel genes and transcripts identified from long-read sequencing is not clearly defined. Low conservation and lack of H3K4me3 signal (fig2c and d) suggests possible

false positives. Some other comparisons against validated genes such as exon count and transcript length could be provided.

2. The parent of origin and allelic specific transcripts analyses (figures 4 and 5) seem redundant. Without having the reciprocal crosses (e.g. C57Bl male x DBA2 female in addition to the cross used in the study) I don't see how you can unambiguously pull these two effects apart.

Detailed comments:

Line 153 and Figure 1. The classification of transcripts as 'potentially novel gene' and 'Other' is not clear and should be described in the main text.

Line 165 and Figure 1. Try to find an alternative classification term to 'else'. Not clear what this means.

Line 202. Try to make the transition from discussing coding to non-coding transcripts clearer.

Line 207 and Figure 2c. Majority of novel non-coding transcripts are NOT conserved. This suggests possible false positives among putative novel genes. For comparison, what are the conservation properties of known non-coding genes?

Line 221 and Figure 2d. The H3K4Me3 signal at novel TSS and Novel gene TSS is weak (or non-existent) compared with 'Classic' TSS's. Again this suggests possible false positives.

What is the Length distributions of novel vs annotated transcripts? Similarly what is the exon count of novel vs annotated transcripts?

The final set of high confidence novel genes / isoforms is not clearly defined. Once it is, it

would be interesting to see some of the potential functional roles and temporal expression patterns of these genes.

Line 235 – 252. This section is hard to follow.

Lines 309-3314. Description of AS identification is redundant with lines 266-271.

Line 316 and Figure 3e "...rare significant DAS events, were retained from 1-cell to blastocyst transitions.....". How is this represented in Figure 3e?

Line 319 and Figure 3f. The GO term analysis does not seem very informative. Consider making this supplementary figure.

Line 331 – Seems like overstatement. These data just demonstrate the presence of AS events in preimplantation embryo development.

Figures 4 and 5. As mentioned above, it is not clear to me how you separate the effects of parent vs strain specific expression with available data. These sections seem quite redundant.

Line 362-366 and Fig 4d,e. The terms 'consistent' and 'inconsistent' are not clear to me. I was unable to interpret these figures. Perhaps some schematic of the analysis would be useful

Response to reviewer #1:

In their manuscript, “High resolution and allelic-specific annotation of preimplantation embryo transcriptome using long-read sequencing”, the anonymous authors carry out transcriptome sequencing of the beginning part of the germinal stage in embryogenesis using PacBio SMRT sequencing. The long-reads SMRT sequencing can produce were leveraged to more fully characterize the mouse preimplantation transcriptome, uncovering many novel putative coding and non-coding transcripts as well as novel putative splicing isoforms. In addition, the authors followed their earlier work (ref 11) on elucidating parent of origin effects to identify allele-specific alternative splicing events where changes in splicing patterns over time as well as levels of expression were demonstrated in an allele-specific way, with suggestions relating an evolving understanding of the zygotic gene activation events that occur as part of early embryogenesis.

Casting a more comprehensive net by employing multiple technologies as was done in this manuscript is critical to advance to a more accurate annotation of a transcriptome, and the authors provide ample evidence that there may exist entirely new aspects of biology that have heretofore gone unobserved. The uncovering of several thousand novel transcripts and novel splice forms in the earliest stages embryogenesis (with the authors reporting > 20% on average of the transcripts detected were novel, a striking result if these all turn out to be validated), along with the demonstration of strong parent-of-origin effects observed in alternative splicing events as well as the dynamics of those events, well demonstrates the potential value of these findings as a general resource that will be of high interest to anybody who thinks about transcriptome data. The question really is how much of the discoveries reported are valid functionally, given no experimental support is given regarding any of the findings playing a functional role, and then given the claims that the long-read sequencing really enabled many of the findings, what role the long-reads played in the general findings reported. For validation, the authors employed what could be a pretty comprehensive set of bioinformatic analyses to support that the novel transcripts and isoforms were likely functional (comprehensive characterization of homology to existing proteins and domains, evolutionary conservation, epigenetic enrichments in the TSS, and so on). But often the details regarding the results are not clearly expressed and confusing at times, and then there is not very strong statistical rigor throughout, so difficult to interpret the true scope of the findings.

Response: Thanks very much for the reviewer #1’s constructive suggestions and helpful comments, and we highly agree with the reviewers’ views. Accordingly, we have made following major revisions: (1) we randomly selected 20 novel transcripts (including 10 novel genes and 10 novel splicing isoforms) with relatively high expression for PCR-Sanger sequencing validation; (2) we disrupted two novel genes (a non-coding RNA and the right coding frame of *Kdm4d*) identified by long-read sequencing using CRISPR/Cas9 system, and revealed its essential functions for early embryogenesis; (3) for allele-specific alternative splicing part, it is really hard to validate its functions by using current technology; considering the suggestions from reviewer #2 and #3 together, we

simplified this part into Figure 5 to propose that long-read SMRT sequencing can help to accurately recognize the origin of parental DNA and to discover the allele-specific alternative splicing as an perfect example; (4) we sincerely apologize for unclear or confusing description in our manuscript, and we have added statistical analysis and carefully revised our manuscript in our revision.

Specific Comments:

1. Regarding the experimental design (as depicted in fig 1a), to help those like myself less familiar with embryogenesis, it would be nice to indicate how far removed the blastocyst is from the earlier stages that were profiled. It seems like a big jump (from 8 cells to blastocyst) and so understanding the rationale for that would be of interest.

Further, it is not clear even how many embryos were characterized at each stage. In the results the authors indicate 150-200 cells were used for each sample, but it is unclear what constitutes a sample. Does this mean that 150-200 zygotes were sequenced? And then 75 2-cell blastomere (excuse my terminology if not completely correct) and so on? Or were the cells batched in order to get enough cells and then moved forward with the prep step? Clearly if batching was done there is the potential to introduce some noise and a loss of resolution. In any case, this should all be clarified.

Response: Thanks very much for the reviewer's constructive suggestions. We are so sorry for presenting Figure 1 improperly, and we have added the depiction for blastocyst with (BL; 32-64C). Actually, blastocyst (32-64 cells) is the following stage of 8-cell embryo (2-3 cell cycles), and these two stages were regularly profiled in our and other's previous papers [1-3] regarding to early mouse embryogenesis.

We are so sorry for the unclear description of embryo or cell numbers used for each sample. We have revised the main text into "Pooled embryos at each stage from the same batch of zygotes, including 150 oocytes (Oo), 150 1-cell embryos (1C), 100 2-cell embryos (2C), 50 4-cell embryos (4C), 25 8-cell embryos (8C), 20 blastocysts (BL, 32-64C), and bulk sperms, were collected for cell lysis", and we added description of embryo numbers in the figure legend for Figure 1A as "For a batch of samples, 150 oocytes (Oo), 150 1-cell embryos (1C), 100 2-cell embryos (2C), 50 4-cell embryos (4C), 25 8-cell embryos (8C), 20 blastocysts (BL, 32-64C), and bulk sperms, were collected for experiments". As for the concerns on batch effect, enough zygotes from one batch were subjected to subsequent development and sample collection, and a complete group of samples were obtained from a same batch. For each batch, every sample were spited into four parts for PCR amplification, and then four parts were mixed together for cDNA purification. To minimize the batch effect, we amplified two batched of cDNA samples separately and mixed together to obtain enough amount of cDNAs for long-read sequencing. At the same time, two complete groups of cDNA samples (only a small fraction: 10 ng) were subjected to short-read sequencing separately as two replicates. This issue has been clarified in our revision as "Pooled embryos at each stage from the

same batch of zygotes, including 150 oocytes (Oo), 150 1-cell embryos (1C), 100 2-cell embryos (2C), 50 4-cell embryos (4C), 25 8-cell embryos (8C), 20 blastocysts (BL, 32-64C), and bulk sperms, were collected for cell lysis. Total RNA samples were precipitated with absolute ethanol, and the dissolved RNAs were divided into 4 aliquots for cDNA amplification as previously described.”

2. The dataset appears to have been generated on slightly older technology, given the authors indicated they did a 4-part split in their amplification, which is no longer required for the IsoSeq assay, and in fact carrying out size-selection prior to library construction could create biases in the transcriptional space due to the physical separation carried out prior to sequencing. I'm not advocating in any way that the experiment should be repeated on the newer tech, but rather think it important to discuss whether this could have created a bias and what the implications of that would be for the results reported, if any.

Response: Thanks very much for the reviewer's helpful comments. We are also regretful for not using the newest Iso-Seq technology. Actually, our method was the newest version when our project started up, and it is also very expensive. Considering the cost of our project, we did not repeat the sequencing with newer technology. As for the size selection, to minimize the effects of library construction, we obtained enough amount of cDNAs (up to 2 ug) for library construction without size selection to reduce biases. According to the reviewer's suggestion, we have added discussion as “In addition, we subjected enough cDNAs to long-read sequencing to avoid size selection and bias generation, while newer Iso-Seq technology may help to produce higher-quality data with a reduced DNA input” in our revision.

3. There is some characterization of the novel transcripts and isoforms provided (such as figure 1 and some supp figures), but what seems noticeably absent are summaries relating to the number of exons in the novel transcripts, the exon lengths, the sequence complexity of the exons, and how those statistics compare for the novel transcripts detectable with only short read data versus those requiring long-read data, and then how these compare to GENCODE transcripts. What are the characteristics of those transcripts that were only identified given the long-read data and why couldn't they be assembled from short-read data? Given figure 2a clearly shows the long-reads are important, why not press to explain exactly why?

This seems to get lost in conveying of other results such as 2e, where a novel gene is shown upstream of Tik2, but it looks like this novel transcript could have been easily detected with short reads? So most of the novelty this way is driven by looking in conditions that have not been explored deeply before?

Response: Thanks very much for the reviewer's constructive comments about more detailed characterization of exons in the novel transcripts. First, we compared the

distributions of exon number and exon length in the novel transcripts identified by long-read data and short-read data, respectively (Supplementary Fig. 2e-f). The corresponding distributions of exons in the annotated transcripts from GENCODE (vM20) were also profiled as background. The novel transcripts of short-read data were assembled by Cufflinks. Compared with GENCODE annotated transcripts, novel transcripts identified with short-read data have higher ratio of mono-exonic transcripts, and have higher ratio of exons with more than 1000 bp. However, novel transcripts identified with long-read data showed similar distributions of exon number and exon length of exons compared with GENCODE annotated transcripts. Then, we analyzed the sequence complexity of exons in the novel transcripts by profiling the basic complexity characteristics and calculating the overlapping ratio with pre-defined low-complexity regions, which were defined by RepeatMasker (Supplementary Fig. 2g-j). We chose the four nucleotide percentages, GC and AT percentages, dinucleotide patterns percentages and Complexity of Markov model values. The first six value of complexity of Markov model values were chose as signatures. Both novel transcripts showed similar complexity characteristics comparing with GENCODE annotated transcripts. Compared to the overlapping ratio of GENCODE annotated transcripts, novel transcripts identified with short-read data have a higher overlapping ratio with low complexity region, while novel transcripts identified with long-read data have a less but closer ratio with low complexity region (Supplementary Fig. 2j).

Supplementary Fig2. (e) The exon count distributions of novel transcripts identified by long-read data, short-read data and transcripts from GENCODE annotation. **(f)** The exon length distributions of novel transcripts identified by long-read data, short-read data and transcripts from GENCODE annotation. **(g)** Average percentage of bases in exons of novel transcripts identified by long-read data, short-read data and transcripts from GENCODE annotation. **(h)** Average percentage of motifs in exons of novel transcripts identified by long-read data, short-read data and transcripts from GENCODE annotation. **(i)** Complexity of Markov model values in in exons of novel transcripts identified by long-read data, short-read data and transcripts from GENCODE annotation. **(j)** We compare the exons with low complexity regions, the histogram showed the overlap proportion of exons of novel transcripts identified by long-read data, short-read data and transcripts from GENCODE annotation.

Furthermore, we profiled the same characteristics of transcripts that can be only identified by long-read data or by short-read data. Compared with transcripts that can only be identified by short-read data, the distributions of exon number, exon length of exons, and the overlapping ratio with low-complexity region in the transcripts that can only be identified by long-read data are more similar to those of GENCODE annotated transcripts (**Supplementary Fig.2k-p**). Together, our results showed that comparing with transcripts identified with short-read data, transcripts identified with long-read data shared more similar characteristics with GENCODE annotated transcripts.

Supplementary Fig2. (k) The exon count distributions of transcripts only identified by long-read data, short-read data and transcripts from GENCODE annotation. **(l)** The exon length distributions of transcripts only identified by long-read data, short-read data and transcripts from GENCODE annotation. **(m)** Average percentage of bases in exons of transcripts only identified by long-read data, short-read data and transcripts from GENCODE annotation. **(n)** Average percentage of motifs in exons of transcripts only identified by long-read data, short-read data and transcripts from GENCODE annotation. **(o)** Complexity of Markov model values in in exons of transcripts only identified by long-read data, short-read data and transcripts from GENCODE annotation. **(p)** We compare the exons with low complexity regions, the histogram showed the overlap proportion of exons of transcripts only identified by long-read data, short-read data and transcripts from GENCODE annotation.

[Redacted]

[Redacted]

4. I like how the authors quantified the isoforms by mapping short reads to the merged isoform dataset across all stages. However, it appears only HQ transcripts were used, but perhaps it would be better to merge the PacBio subreads and run IsoSeq on the merged raw data? This would likely increase the representation of isoforms/rescue isoforms that likely dropped out in the cluster/refine steps when processed separately.

Response: Thanks very much for the reviewer's helpful comments. We used cuffquant from cufflinks suite and treated short-read alignments at each stage as input data, to quantify the expression reads of merged isoforms in each stage separately.

At the beginning of our data analysis, we have tried two methods to obtain merged transcripts in 8-cell and blastocyst data: run separated and merge later; merge data and run later.

[Redacted]

5. The analysis tools that were used (Cuffcompare, SUPPA, StringTie, CPAT, ect.) all seem fine as I think they had been used in a paper I had always considered as a gold standard for novel transcript annotation:

However, today there are what could be considered as improved packages were released:

<https://www.ncbi.nlm.nih.gov/pmc/articles/PMC4417758/>

and now streamlined into one tool, SQANTI2. I'm sure the authors are familiar with this, and I again am not advocating that they redo everything with these improved tools, but would be interested in whether the authors think these newer tools may provide a more comprehensive and accurate view.

Response: Thanks very much for the reviewer's kind reminding and understanding. As suggested by the reviewer, we applied SQANTI2 on our filtered full-length transcripts.

[Redacted]

6. *Figure 2d is difficult to interpret. The legend or methods/supp do not really explain exactly what was done, what the enrichments represent, how to assess what is statistically significant and so on. For example, in the “novel gene TSS” result the*

H3K4me3 enrichment pattern does not seem to be well correlated with expression. Is that expected? Further, the correlation for the classic TSS category doesn't seem as strong as for the novel TSS, which doesn't seem right. What about for annotated TSS's? Was this supposed to be represented in supp figure 4? Basically, this should all be clarified more quantitatively with statistical significances as well, conveying what actually is expected by chance versus what is expected as the gold standard.

Response: Thanks very much for the reviewer's constructive comments. We have revised our legends carefully to make it clearer for this part. As the reviewer mentioned, the gene expression and H3K4me3 enrichment was not correlated well. From our point of view, it was expected and reasonable, and there were three main reasons: (1) early embryogenesis is very special for histone modifications and H3K4me3 was re-established after fertilization [6] and the global enrichment of H3K4me3 at 2-cell stage was very low as a transition stage (Figure S4a); however, the expression of genes includes a large percentage of parental mRNAs which were not controlled by H3K4me3 enrichment; (2) for a single gene, the expression of genes were generally correlated with H3K4me3 enrichment, while for different genes, the correlation between expression and H3K4me3 enrichment was not very well, because different genes were enriched with H3K4me3 signals with differential range of modifications and different basal levels; (3) here we used the H3K4me3 enrichment only to prove the reliability of our identified novel transcripts, because H3K4me3 is a pre-condition for gene expression/activation. Especially for novel isoforms of annotated genes with novel TSSs, there is no other standard to judge the reliability of these transcripts. Here we observed that the transcription start sites of these transcripts were enriched with apparent H3K4me3 peaks with peak-valley-peak signals; this evidence largely increased the confidence of these kind of novel transcripts (Figure 2d, middle line). Finally, the gold standard for H3K4me3 enrichment is peak-valley-peak enrichment in gene promoters (as shown in Figure S4a) and the enrichment of H3K4me3 enrichment was calculated by deducting the background from input signals using MACS2 with default parameter -g mm -B -q 0.05 --nomodel --broad --shiftsize=73 -SPMR;

7. For the GO enrichments such as that depicted in Supp figure 3a, it is unclear how the enrichments were computed, how novel genes were assigned to GO categories. This doesn't appear to be described anywhere in the materials provided. Do the novel genes take on categories that relate to their closest homolog? Or gene with closest sequence identity? Or???

Response: Thanks very much for the reviewer's helpful comments. We sincerely apologize for the lack of explanation. To infer the potential function of novel transcripts, we searched the homology protein of the novel coding transcripts and assigned the novel coding transcripts with these proteins using BLASTP. The GO enrichment analysis was performed on the set of the homology proteins using STRING. We revised the description of the GO enrichments of Supplementary Figure 3a as "The GO enrichment analysis in biological processes for novel coding transcripts predicted by annotated proteins.", and

also revised the corresponding method part to describe the detailed method as “In addition, we performed GO enrichment analysis in biological processes using STRING (v11.0) 13 for the set of known homology proteins corresponding to the novel coding transcripts. The homology proteins were defined as known proteins corresponding to the ORFs of novel coding transcripts with ident=100 in Uniprot” in supplementary methods.

8. In several locations the authors use adjectives such as giant, remarkable, and drastic to describe results, but these are never associated with any statistical tests that would help the reader understand what hypotheses are being tested and why the result reported should be deserved of such adjectives. For example, line 244 in the main text indicates, “...novel isoforms showed giant mapping read change...”, with a delta statistic given, but no description on why the statistic given is giant, what would be expected by chance and how significant the observed result actually is.

Another example is line 302, where the authors indicate that the expression of several splicing factors were “remarkably upregulated during 1-cell to 2-cell transition” with reference to supp figure 7h, but this is not really well shown in the figure, given the color scale given for the z-score used to depicted the expression ranged for -2 to 2, so that the max color under the assumption of the values being normally distributed would be only nominally significant and not significant if one were to correct for multiple testing. But no description of statistical significance, false discovery rates or other such characterizations are provided, making it difficult to understand if the results presented are truly “remarkable”.

The same goes for line 315 in which the authors state, “River plot showed drastic global changes of significant DAS events during embryonic development.” Where in fact it is difficult to understand what exactly the river plot (figure 3e) shows in this instance. There are no statistical significances given for the up or down regulation events shown, no significance for the changes in these regulation patterns over the different stages, and then nothing indicated regarding any kind of correction needed for the expansive multiple testing that is happening in these data. None of this is well described in any of the methods material.

Response: Thanks very much for the reviewer’s kind reminding. We agree with the reviewer that we used many adjectives in our description, and we revised our manuscript with soft tongue throughout the main text. Moreover, we added detailed description of statistical analysis in figure legend or methods for all needed data. For mentioned examples:

In mentioned line 244, we profiled the number of mapping reads using two different transcriptomes, and used circos plot to illustrate transcripts with different mapping reads in two transcriptomes. We applied Wilcoxon matched-pairs signed rank-sum test to

evaluate the significance of the difference between mapping reads in two transcriptome, and p -value is smaller than 1×10^{-100} .

For the significance of the expression changes of splicing factors, we applied the z-score normalization on the expression matrix of splicing factors to emphasize the difference between stages. To precisely address the significance of differential expression of splicing factors between stages, we used DESeq2 to identify differentially expressed splicing factors. Significantly differentially expressed splicing factors were defined as splicing factors with $p\text{-adj} < 0.05$ (Supplementary Table 5). To make it more accurate, we revised the Supplementary Figure 8h and only marked the differential expressed splicing factors defined by DESeq2 between 1cell and 2cell.

In the river plot of Figure 3e, we profiled dynamic changes of master list of all differential alternative splicing events. The “significant” means the differential alternative splicing (DASs) events were identified as significantly different by SUPPA2. The “up-regulation” and “down-regulation” was decided by the ΔPSI calculated by SUPPA2. The ΔPSI and corresponding P-value of all DAS events were already provided in Supplementary Table 6. To illustrate the significant changes of differential alternative splicing events, we performed Fisher-exact test on the number of DAS events which is retained during two groups according to the reviewer’s constructive comments. The results in Supplementary Table 7 support that the DAS events were significantly changed during the embryonic development.

Supplementary Table 7.

	1cell-2cell→2cell-4cell	2cell-4cell→4cell-8cell	4cell-8cell→8cell-blastocyst
up	0.0003587	1.87E-34	3.72E-17
down	0.0001029	1.52E-12	1.52E-12

9. Further regarding statistical significance, in results such as that depicted in figure 3f, corrected p -values should be shown or false discovery rates so that the reader can understand what was actually expected by chance. Further, for enrichment statistics such as those provided, the “effect sizes” should be given as well (in this case would be a fold-enrichment or odds ratio) since the p values will be correlated with sample size and the very large and broad categories will be detected with higher power and will be less relevant since they really don’t tie you into any specific biology.

Response: Thanks very much for the reviewer’s important reminding. The concerns about the corrected p -value of GO analysis is very important. Accordingly, we provide the detailed output of the GO analysis in a new supplementary sheet, including fold enrichment and FDR (Supplementary Table 8).

10. In lines 327-333 the authors nicely give some PCR validation of a single novel and two novel isoforms. However, these appear to have been selected to highlight out of

thousands of novel genes/isoforms detected, but no justification is given as to why these 3 examples were picked for validation. How many validations were attempted? Was there any prioritization given to the examples that were selected?

Response: Thanks very much for the reviewer's helpful comments. We apologize for lack of describing the criteria for selecting genes for validation. To further strengthen this part, we selected 10 novel isoforms of annotated genes and 10 novel genes, which showed apparent expression in RNA-seq data for these novel transcripts at least in one stage, for PCR validation. All of these transcripts were successfully PCR-amplified and validated by Sanger sequencing (**Supplementary Figure 5**).

More General Comments:

1. I have not made any comments relating to grammar to include coherent sentence structure, but significant attention should be paid to significantly smoothing out the text throughout. As an example, in the first sentence the reader encounters (the first sentence of the summary paragraph), the sentence reads, "The development of third generation of long-read sequencing...", which should be corrected to something like, "The development of third generation, long-read sequencing technologies..." or something like that. Grammatical errors, spelling mistakes, and confusing sentence structures occur throughout.

Response: Thanks very much for the reviewer's helpful comments. We really apologize for improper description of some sentences. We have carefully revised our manuscript and send our paper to American Journal Experts recommended by the journal for native English editing.

2. For the supplementary material provided as excel spreadsheets, there is no legend or description given, making it nearly impossible to understand what is actually provided, what the different columns represent, how the values were computed and so on. It's clearly a lot of material that would be very useful to have described. Perhaps an issue in the conversion for submission (I got the excel spreadsheets, the supp tables, from the merged zip file that was provided)?

Response: Thanks very much for the reviewer's kind reminding. We have to apologize for the absence of the legend and description for the supplementary materials. In our revision, we added detailed table legends to all supplementary materials in the supplementary files and submitted these files directly as supplemental materials.

3. Related to the above lack of description around the supplementary material, I did not see anywhere described where the data used to derive all of the results reside. As it

stands it is not possible to reproduce any of the results presented without access to those data.

Response: Thanks very much for the reviewer's kind reminding. Your concerning about the data availability is very important. We submitted all the sequencing data including long-read data, short-read data, and transcriptome annotation in gtf format and expression matrix. The GEO number was already provided in the manuscript as GSE138760. Currently all data are set as private, but you can access with the visiting code kjuhegsejjmdnyt. We will set the data public immediately when our study is published.

Response to reviewer #2:

1. Far fewer numbers should be used in the text. It should be clear how many new transcript isoforms and novel genes were detected, but the constant comparison and percentages of various details is confusing and misleading. This is confounded by the fact that none of the numbers seem to add up. For example, in Figure 1b,c, the proportion of orange and light blue bars does not correspond even in the slightest, and what does "Sequential subset of exons contained within annotation" mean? The description and depiction in Fig. 1a is very clear and understandable, and hence it would make sense to stick to those same 3 groups ("known isoforms", "novel isoforms", "novel genes") throughout. If "other" refers to sequencing errors or other artificial reads that have no biological relevance, they should be eliminated during the first steps of the analysis and not displayed.

Within these many comparisons, the authors conclude that almost half of their novel genes are long non-coding RNAs. Does this relate only to the "novel gene" group, or to all novel isoforms & genes? The fraction seems too high.

Response: Thanks very much for the reviewer's helpful comments and we sincerely apologize for unclear description of specific number of categorized transcripts as well as the definition of some categories. In our revision, we added description for results with more specific numbers. According to the limitation of sequencing or computational methods, we have to first annotate all detected transcripts and compare them with annotated transcripts (as described in Figure 1a as "Transcription annotation-Cuffcompare"). As for the categories of transcripts in Figure 1b, we have referred this from a recently published paper [7]. We also added description for "Sequential subset of exons contained within annotation" as "reference-annotated exonic subsets of transcripts with missing exons were defined as "sequential subset of exons contained within annotation" and "other" as "including some chromosome-crossed and genome-unrelated sequences, which cannot be classified by the current software and might be generated during the sample preparation or sequencing process, were excluded in subsequent analysis". Considering a part of sequencing data for the whole project, we retained the percentage of "Other" in Figure 1b and 1c, and it was already eliminated in the subsequent analysis.

As the reviewer mentioned, the two classes, "Sequential subset of exons contained within annotation" and "Potentially novel isoform", were not consistent between Figure 1b and c. In Figure 1b, we presented the annotation results of transcripts of each stage separately. In Figure 1c, we presented the annotation results of transcripts merged from seven stages. We used Cuffmerge to merge the transcripts, and the number of transcripts was collapsed or merged in this process, especially for annotated genes. Thus, the percentage of each class in merged transcripts was different from the annotation of transcripts of each stage.

In Figure 1e, we demonstrated that the fraction of novel transcripts based on coding potential calculated by CPAT. The novel transcripts in Figure 1e include the “novel genes” and “novel isoforms” in Figure 1a. The percentage of the long non-coding RNAs was consistent with previous study [8], possibly because lncRNAs were more difficult to be identified by short-read sequencing and assembly.

2. *The oocyte transcriptome has been custom-annotated before (Gahurova et al., Epigenetics & Chromatin 2017). The authors must compare their data to this previous annotation update, in particular with regards to the often-unusual exon usage in oocytes.*

Response: Thanks very much for the reviewer’s helpful suggestions about the comparison with published oocytes transcriptome. Gahurova et al. and their previous study (Deep sequencing and de novo assembly of the mouse oocyte transcriptome define the contribution of transcription to the DNA methylation landscape) annotated the oocyte transcriptome using short-read data with 5,877 oocyte cells at four stages. Using the short-read data of large scale, they identified 82,939 transcripts forming 39,099 expressed genes. Our merged short-read and long-read transcripts identified fewer number of transcripts (15,271 transcripts v.s. 82,939 transcripts and 10,190 genes v.s. 39,099 genes), which is due to the limitation of sequencing depth of third-generation sequencing, which is comparable with another study with Pacbio sequencing [7].

[Redacted]

3. *It is essential that solid verifications are performed to confirm the existence of novel candidate genes/isoforms, I suggest in the order of magnitude of at least 10 each. The very few examples shown in Fig. 3h are not sufficient evidence by any stretch. The bands shown are unconvincing (e.g., Ypel5) and will need to be confirmed by Sanger sequencing to indeed reflect the candidate transcript. Similarly for the XLOC example, the 750 bp band would normally be considered as unspecific and will need sequence-confirmation. If these are the best examples of confirmatory PCRs, they casts significant doubt on the data reported. If indeed the sequencing data suggest the concomitant presence of multiple transcript isoforms in the same cell/stage in different proportions, then the authors should indicate estimated proportions (from the sequencing data) and still confirm the obtained PCR bands independently by Sanger sequencing.*

Response: Thanks very much for the reviewer's helpful suggestions. Accordingly, we randomly selected additional 10 novel isoforms of annotated genes and 10 novel genes, which showed apparent expression reads in RNA-seq data for these novel transcripts at least in one stage, for PCR validation. All of these transcripts were successfully PCR-amplified and validated by Sanger sequencing (Supplementary Figure 5).

4. *Figures 4 and 5 are very difficult to follow and should be reworked for being more intuitively comprehensible. Similar to the above comment, splicing isoforms need to be robustly confirmed in individual RT-PCR and Sanger sequencing approaches.*

Response: Thanks very much for the reviewer's helpful comments. Combining the suggestions from three reviewers, we added functional investigation for two genes identified by Pacbio-seq in early embryo development (revised Figure 4) and shortened Figure 4 and 5 into Figure 5 to claim that long-read sequencing technology is helpful for accurately recognizing parent-of-origin transcripts. We have tried to find several examples for PCR validation. However, for those SNPs existed within a short range, it is easily discovered by both long-read and short-read data. For those SNPs existed in a long range that can be only identified by long-read reads, the assembly-based RNA-data cannot obtain allelic specific isoforms using current assembly methods (Figure 5e-f). We have tried to PCR amplify example genes but we failed to sequence a full length of cDNA containing two far SNPs, because the contents containing several kinds of cDNA products containing SNPs.

5. *The parent-of-origin section is not meaningful because the data only originate from one particular cross, C57BL/6 mother x DBA/2 father. Thus, beyond the sperm and oocyte stage and in particular with ZGA, strain-specific differences will have a major effect on allelic biases. To make this section of value, the reciprocal cross would need to be analysed as well. If the reciprocal cross is not performed, this part of the analysis should be removed or significantly shortened, and the caveat of strain- over parent-of-origin-induced biases discussed.*

Response: Thanks very much for the reviewer's helpful comments. We agree with the reviewer that the parent-of-origin section is not meaningful and is hard to validate its functional significance. As mentioned above, combining the constructive suggestions from three reviewers, we added functional investigation for two genes identified by Pacbio-seq in early embryo development and shortened Figure 4 and 5 into one figure to claim that long-read sequencing technology is helpful for accurately recognizing parent-of-origin transcripts/splicing, and this method can be expanded to study parent-of-origin DNA methylation and imprinting. We agree with the reviewer that the reciprocal cross will be a good strategy to explore the significance of allelic biased splicing. As far as we know, similar work with third-generation sequencing in early embryogenesis was conducted from three labs, and the limited time does not allow us to prepare reciprocal cross samples and sequencing data, which needs at least three months. Combining the constructive suggestions from three reviewers, we added functional investigation for two genes identified by Pacbio-seq in early embryo development (Figure 4) and shortened Figure 4 and 5 into one figure (Figure 5) to prove that long-read sequencing technology is helpful for accurately recognizing parent-of-origin transcripts/splicing than just using short-read sequencing data. As we focused on the advantage of third-generation sequencing in identification of allele-specific transcripts/splicing in our revised manuscript, we removed

the sentences regarding parent-of-origin effects. Therefore, please forgive us that we did not add the discussion of the caveat of strain over parent-of-origin induced biases in revision. Thanks very much for your kind understanding.

Response to reviewer #3:

1. The final set of novel genes and transcripts identified from long-read sequencing is not clearly defined. Low conservation and lack of H3K4me3 signal (fig2c and d) suggests possible false positives. Some other comparisons against validated genes such as exon count and transcript length could be provided.

Response: Thanks very much for the reviewer's helpful suggestions. Accordingly, we made more comparison with the annotated transcripts to show the reliability of our transcripts. We first compared the transcript length of novel transcripts with annotated transcripts (Supplementary Fig. 2a). Generally, the distribution of transcripts with differential length for novel transcripts was comparable with that for annotated transcripts. Then, we compared the exon number between novel transcripts and annotated transcripts (Supplementary Fig. 2b). The distributions of exon number between novel and annotated transcripts were similar, but annotated transcripts contained more transcripts consisted of more than 10 exons. Together, these data demonstrated that our identified novel transcripts possess similar characteristics compared with annotated transcripts. To further strengthen our data, we selected 10 novel isoforms of annotated genes and 10 novel genes, which showed apparent expression in RNA-seq data for these novel transcripts at least in one stage, for PCR validation. All of these transcripts were successfully PCR-amplified and validated by Sanger sequencing (Supplementary Figure 5).

Supplementary Fig2. (a) The length distributions of annotated and novel transcripts. (b) The exon count distributions of annotated and novel transcripts.

2. The parent of origin and allelic specific transcripts analyses (figures 4 and 5) seem redundant. Without having the reciprocal crosses (e.g. C57Bl male x DBA2 female in addition to the cross used in the study) I don't see how you can unambiguously pull these two effects apart.

Response: Thanks very much for the reviewer's helpful comments. We agree with the reviewer that the parent-of-origin section is redundant and is hard to validate its functional significance. Combining the constructive suggestions from three reviewers, we added functional investigation for two genes identified by Pacbio-seq in early embryo development (Figure 4) and shortened Figure 4 and 5 into one figure (Figure 5) to prove that long-read sequencing technology is helpful for accurately recognizing parent-of-origin transcripts/splicing than just using short-read sequencing data. We also agree with the reviewer that the reciprocal cross will be a good strategy to explore the significance of allelic specific transcripts/splicing. As far as we know, similar work with third-generation sequencing in early embryogenesis was conducted from three labs, and the limited time does not allow us to prepare reciprocal cross samples and sequencing data, which needs at least three months. Thanks very much for your kind understanding.

3. Line 153 and Figure 1. The classification of transcripts as 'potentially novel gene' and 'Other' is not clear and should be described in the main text.

Response: Thanks very much for the reviewer's kind reminding. We should apologize for the absence of the detailed definition in transcriptome annotation. Our transcriptome annotation was based on the class code from Cufflinks and a previous study [7]. In our revision, we have added description for the terms 'potentially novel gene' and 'Other' as below.

"The novel transcripts, including "potentially novel genes" (no overlapping with the reference annotation) and "potentially novel isoforms"

Specifically, Class code "=" was described as "Exact match to annotation"; class code "c" was described as "Sequential subset of exons contained within annotation"; class code "e" and "j" were described as "Potentially novel isoform"; class code "u" was described as "Potentially novel gene"; class code "i", "o", "p", "r", "x", "s" were described as "Other". In Cufflinks definition of each class code, "u" refers to "Unknown, intergenic transcript".

[Redacted]

4. Line 165 and Figure 1. Try to find an alternative classification term to 'else'. Not clear what this means.

Response: We sincerely apologize for not describing the term “else”, and we have added description for this term in revision as below.

“Next, the merged transcripts, including the annotated transcripts and the novel transcripts (including potentially novel isoforms and novel genes), were further classified into protein-coding transcripts, long noncoding RNAs (lncRNAs), and else (not belonging to protein coding RNAs or lncRNAs, such as processed pseudogenes, snoRNAs, and rRNAs) according to the GENCODE annotation or protein-coding potential of the novel transcripts using CPAT (Coding-Potential Assessment Tool, v2.2.0).”

5. Line 202. Try to make the transition from discussing coding to non-coding transcripts clearer.

Response: Thanks very much for the reviewer’s helpful comments. We added a sentence “Compared with the fully explored protein-coding regions of the whole genome, the functions of non-coding sequences, which constitute over 98% of the genome, remain largely unknown.” before our discussion of non-coding transcripts.

6. Line 207 and Figure 2c. Majority of novel non-coding transcripts are NOT conserved. This suggests possible false positives among putative novel genes. For comparison, what are the conservation properties of known non-coding genes?

Response: Thanks very much for the reviewer’s helpful comments. In our manuscript, we provided two types of conservation scores of novel non-coding transcripts compared with the random control regions in **Supplementary Figure 3c-d**. The results showed that the novel non-coding transcripts were more conserved than random control regions.

[Redacted]

7. Line 221 and Figure 2d. The H3K4Me3 signal at novel TSS and Novel gene TSS is weak (or non-existent) compared with 'Classic' TSS's. Again this suggests possible false positives.

Response: Thanks very much for the reviewer's helpful comments. As previously reports [7, 9], we have performed conservation and coding potential analysis for novel transcripts. We first used histone modification dataset to strengthen the reliability of our newly identified transcripts. For those novel transcript with weak or low H3K4me3 signals, there might be two reasons: (1) early embryogenesis is very special for histone modifications and H3K4me3 was re-established after fertilization [6] and the global enrichment of H3K4me3 at 2-cell stage was very low as a transition stage (Figure S4a); however, the expression of genes includes a large percentage of parental mRNAs (especially maternal mRNAs) which were not controlled by H3K4me3 enrichment at that stage; (2) these novel transcripts might be expressed with a relatively low levels relative to previously annotated isoforms, which results in hard identification with short-read data. For those high-confidence novel transcripts, the TSSs of these transcripts were enriched by H3K4me3 with classic peak-valley-peak signals (Supplementary Figure 4). Moreover, we selected 10 novel isoforms of annotated genes and 10 novel genes for PCR validation, and all of these transcripts were successfully PCR-amplified and validated by Sanger sequencing (Supplementary Figure 5).

8. What is the Length distributions of novel vs annotated transcripts? Similarly what is the exon count of novel vs annotated transcripts?

Response: Thanks very much for the reviewer's helpful comments. Your suggestion of more comparison between novel and annotated transcripts is very important. We have made comparisons for length distributions and exon counts of novel vs annotated transcripts (as shown above in Supplementary Fig. 2a-b).

9. The final set of high confidence novel genes / isoforms is not clearly defined. Once it is, it would be interesting to see some of the potential functional roles and temporal expression patterns of these genes.

Line 235 – 252. This section is hard to follow.

Response: Thanks very much for the reviewer's helpful comments. As shown in Figure 2d, the H3K4me3 enrichment of all novel transcripts were mapped to their transcription start sites, generally showing the enrichment of H3K4me3 enrichment with peak-valley-peak signals. To look into the detail, considering the peak extension of H3K4me3 peaks, we counted the novel transcripts with apparent H3K4me3 peaks within 300 bp of identified newly TSSs. Therefore, we defined those novel transcripts with apparent H3K4me3 peaks within 300 bp of identified newly TSSs as "high confidence novel genes / isoforms". According to the reviewer's suggestion, we have identified two functional transcripts (a novel non-coding gene and an updated coding frame of *Kdm4d* gene) involved in early embryogenesis (Figure 4).

In addition, we revised mentioned line 235-252 to describe how the augmented transcriptome annotation can help improve the accuracy of the quantification of the transcript expression. The revised paragraph was attached below.

"By combining these novel transcripts, we augmented the GENCODE annotation with the novel transcripts derived from merged transcripts and quantified the mapping reads of the GENCODE-annotated transcriptome and augmented transcriptome using salmon (v0.10.0). We observed that nearly half of GENCODE transcripts possessing potential novel splicing isoforms (11,516 out of 23,970, 48%) showed large mapping read changes ($\delta > 5$) for the newly augmented transcriptome references ($P < 1 \times 10^{-100}$, Wilcoxon matched-pairs signed rank test; Fig. 2f). Compared with the expression of the GENCODE-annotated transcripts, the expression of the newly identified transcripts also showed stage-specific expression patterns across preimplantation development (Supplementary Fig. 4e-f), and the expression of these novel transcripts can be integrated to generate a more complete transcriptome landscape. Thus, we identified a large number of novel long-read transcripts, verified thousands of high-confidence novel isoforms and genes, and improved the annotation of preimplantation transcriptome."

10. Lines 309-314. Description of AS identification is redundant with lines 266-271.

Response: Thanks very much for the reviewer's helpful comments. In mentioned line 309-314, we described the identification results of the differential alternative splicing events between two stages using the merged transcriptome. And the alternative splicing events described in mentioned lines 266-271 were alternative splicing event defined based on the specific transcriptome for each stage. We are so sorry for our unclear description about the description in mentioned lines 309-314 (revised line 358-361) and 266-271 (revised line 312-317).

11. Line 316 and Figure 3e "...rare significant DAS events, were retained from 1-cell to blastocyst transitions.....". How is this represented in Figure 3e?

Response: Thanks very much for the reviewer's helpful comments. We have to apologize for the poor presentation in mentioned lines 316 and Figure 3e. In Figure 3e, we profiled the master list of differential alternative splicing events between stages using river plot. The three groups in Figure 3e referred to "up-regulated differential alternative splicing events", "down-regulated differential alternative splicing events" and "not significantly differential". The "up-regulated" means the Δ PSI calculated by SUPPA2 was positive, while "down-regulated" means the Δ PSI value was negative. Figure 3e was designed to show that most of the differential alternative splicing events were not retained from 1-cell to blastocyst stages. None of the differential alternative splicing events can be retained as "up" or "down" during the whole development process. Thus, we made the conclusion that "rare significant DAS events were retained from 1-cell to blastocyst transitions".

12. Line 319 and Figure 3f. The GO term analysis does not seem very informative. Consider making this supplementary figure.

Response: Thanks very much for your concerning for the GO term analysis in Figure 3f. According to the reviewer's suggestion, we moved the mentioned data into **Supplementary Figure 9h**.

13. Line 331 – Seems like overstatement. These data just demonstrate the presence of AS events in preimplantation embryo development.

Response: Thanks very much for the reviewer's kind reminding. We should apologize for our overstatement of our results. We revised the mentioned sentence into "Taken together, these data demonstrate the potential relationship between stage-specific splicing factors and the dynamic changes of alternative splicing events during the preimplantation embryo development." **to demonstrate the presence of AS events in early embryogenesis.**

14. Figures 4 and 5. As mentioned above, it is not clear to me how you separate the effects of parent vs strain specific expression with available data. These sections seem quite redundant.

Response: Thanks very much for the reviewer's helpful comments. We agree with the reviewer that the parent-of-origin section is redundant. Combining the constructive suggestions from three reviewers, we shortened Figure 4 and 5 into one figure (Figure 5) to prove that long-read sequencing technology is helpful for accurately recognizing parent-of-origin transcripts/splicing than just using short-read sequencing data.

15. Line 362-366 and Fig 4d,e. The terms 'consistent' and 'inconsistent' are not clear to me. I was unable to interpret these figures. Perhaps some schematic of the analysis would be useful.

Response: Thanks very much for the reviewer's kind reminding. We apologize for unclear description for the mentioned issue. In detail, the term "consistent" in previously Figure 4d-e means "the expression patterns were highly correlated ($r^2 > 0.1$ and $p\text{-value} < 0.05$)", and the term "inconsistent" in previously Figure 4d-e means "the expression patterns were not correlated ($r^2 < 0.1$ or $p\text{-value} > 0.05$)". Indeed, this definition is a little obscure. According to the reviewers' suggestions, we have shortened Figure 4 and 5 into one figure (Figure 5), and this part was removed in our revision to make our work cleaner.

References for response letter

1. Du, Z., et al., *Allelic reprogramming of 3D chromatin architecture during early mammalian development*. Nature, 2017. **547**(7662): p. 232-235.
2. Zhang, B., et al., *Allelic reprogramming of the histone modification H3K4me3 in early mammalian development*. Nature, 2016. **537**(7621): p. 553-557.
3. Wang, L., et al., *Programming and inheritance of parental DNA methylomes in mammals*. Cell, 2014. **157**(4): p. 979-991.
4. Chen, J., et al., *Spatial transcriptomic analysis of cryosectioned tissue samples with Geo-seq*. Nat Protoc, 2017. **12**(3): p. 566-580.
5. Byrne, A., et al., *Realizing the potential of full-length transcriptome sequencing*. Philos Trans R Soc Lond B Biol Sci, 2019. **374**(1786): p. 20190097.
6. Liu, X., et al., *Distinct features of H3K4me3 and H3K27me3 chromatin domains in pre-implantation embryos*. Nature, 2016. **537**(7621): p. 558-562.

7. Nudelman, G., et al., *High resolution annotation of zebrafish transcriptome using long-read sequencing*. Genome Res, 2018. **28**(9): p. 1415-1425.
8. Casero, D., et al., *Long non-coding RNA profiling of human lymphoid progenitor cells reveals transcriptional divergence of B cell and T cell lineages*. Nat Immunol, 2015. **16**(12): p. 1282-91.
9. Chen, H., et al., *Long-Read RNA Sequencing Identifies Alternative Splice Variants in Hepatocellular Carcinoma and Tumor-Specific Isoforms*. Hepatology, 2019. **70**(3): p. 1011-1025.
10. Patro, R., et al., *Salmon provides fast and bias-aware quantification of transcript expression*. Nat Methods, 2017. **14**(4): p. 417-419.

Reviewers' Comments:

Reviewer #1:

Remarks to the Author:

The authors have gone to very significant lengths to address all reviewer concerns. I am satisfied with the revised manuscript and think it is a solid piece of work that will be appreciated by the research community.

Reviewer #2:

Remarks to the Author:

The manuscript is much improved, and overall the authors have done a great job in addressing the reviewers' comments.

Amendments that should still be introduced:

I agree with the fellow reviewer that the H3K4me3 enrichment at novel TSSs is very weak and not very convincing implying the discovery of false-positives. A comparison with CAGE data would be most helpful.

The allele-specific section is improved by being shortened, I would still argue it should be deleted entirely because it is meaningless without the reciprocal cross. This would not take anything away from the overall strength of this manuscript, in particular now that the authors include KO data of two candidate genes.

Line 132 ff: "...we profiled the transcriptome using both long- and short-read sequencing across mouse preimplantation developmental stages in female C57BL/6J and male DBA/2 inbred mice".

This presumably is meant to read, "in embryos derived from C57BL/6J (female) x DBA/2 (male) crosses." Please correct.

Lines 157-159, it is still unclear what "sequential subset of exons contained within annotation" means, and how it is different from novel isoform. A diagrammatic depiction of these categories in the supplement is needed.

Line 184: only 54.7% of novel [pls insert] transcripts

Lines 250-253: "Among 6,289 potentially novel isoforms, 4,273 transcripts possessed classic TSSs overlapping with the TSSs of previously annotated isoforms, while 2,014 novel isoforms [not TSS] possessed potentially novel TSSs; among the potentially novel genes, 2,266 transcripts might have potentially novel TSSs.

Please change TSS to novel transcripts (if my interpretation of this sentence is correct).

Line 297: It is not the case that “thousands” of novel genes and transcript isoforms were “verified”. Only some 10 were verified. Please rephrase.

Lines 344ff and 350ff: Too many confusing abbreviations, please delete SF and DAS and leave the words spelled out.

Fig. 4G, please provide the numbers of embryos scored. The significance values appear too marginal for the large discrepancies of the bars; please also provide primary data of these counts.

Myriam Hemberger

Reviewer #3:

Remarks to the Author:

This manuscript describes long-read sequencing of the transcriptome of the pre-implantation embryo. The manuscript has been improved with this revision, but further work is required to make it more easily understandable. The technical description of the transcriptome dataset is extensive and the data is likely to be a useful resource for studies of the pre-implantation embryo. The novelty of the data is hard to assess as there is currently no comparison with the state-of-the-art knowledge of the early embryonic transcriptome. The manuscript could be improved with more exploration or demonstration of the biological significance of the findings.

Fig 1. High-resolution annotation of the preimplantation mouse embryo transcriptome with a combination of long- and short-read sequencing

Overall in this section I'd be interested in how the data compares with the state of the art knowledge of pre-implantation embryo transcriptomes. This was raised by another reviewer and was addressed in reviewers response, but there is still no description in the main text. Figure 1A is helpful, but does not seem to precisely match the workflow as described in the text.

Line 138-146. The batching strategy remains unclear.

Line 138. Should this be: “For each stage Total RNA samples were precipitated...”?

Line 142. The samples subject to short read sequencing, and the purpose of short read sequencing is not clearly stated.

Fig 1B and C. It is confusing to use two different plot formats to show the same information.

Line 169. Should this be: “revealing 2,280 transcripts from potential novel genes”?

Line 171. What is the number of full length PacBio reads supporting novel transcripts / genes compared with annotated transcripts? This would also be good validation to have.

Fig 1D/E How do these categories align with those in figure C? Does novel transcripts include both novel gene and novel isoform?

Line 189. Can you speculate why the fraction of coding vs non-coding is so different for

annotated vs novel transcripts? Perhaps the comparison should be done against a more comprehensive reference annotation database (e.g. including <http://www.noncode.org/>)

Line 192. Here we are introduced to transcripts identified from short-read data, but there is no description in the main text (or in fig 1A) of where these come from.

Fig 1f. As for fig1 d,e, what is the definition of novel transcript? It could be that much of the short read mapping signal may be coming from regions that are shared with known transcripts.

Lines 190-213. The comparison with short read data could be described more succinctly.

Fig 2. Validation of novel long-read transcripts in mouse preimplantation embryos

Line 217 and Fig 2a. There is no explanation of what short read plus long read actually means. Was this some kind of co-assembly or merging of transcriptomes. How was this done? What about long-read only?

Line 235. New paragraph for section on non-coding sequences

Line 238, Fig 2C. Is there any complementary analysis of signatures of lncRNA that can be done here

Line 263. New paragraph "To identify high-confidence.....".

What is the coding / non-coding breakdown of the high confidence novel TSSs.

Line 284-296 and Fig 2F. I don't understand the analysis or the figure.

Line 298. The statement "we.... improved the annotation of preimplantation transcriptome" suggests a comparison with prior knowledge of the preimplantation transcriptome. This was not done.

Fig 3. Characterization of alternative splicing in mouse preimplantation embryos

Lines 333-339 and Fig 3d. I'm not sure I understand this. Does 'gain' and 'loss' refer to presence / absence of a transcript isoform? Or is this a quantitative comparison (e.g. 73% transcript isoforms at the 2C stage were more highly expressed than at the 1C stage). Either way this needs clarifying.

Line 343. New paragraph "To explore the potential factors.....".

Lines 351-353. This is a bit tangential but also interesting. Does the correlation between SF expression and alternative splicing make sense in the context of the known modes of action of any of the SFs?

Fig 3e. How does this relate to fig 3d? Does 'down' mean the same thing as 'loss', and up the same as 'gain'? In general this plot is hard to interpret.

Line 365. I don't understand the sentence "Intriguingly, rare significant DAS events were retained from the 1-cell to blastocyst transitions".

It might be useful to show genome browser shots of novel DAS for a few 'famous' embryonic development genes.

Line 372. New paragraph "To identify the potential novel transcripts involved in ZGA...."

Fig 4. Identification of two novel transcripts functionally involved in early embryogenesis

What are the exact genomic coordinates of the regions deleted?

Fig 5. Allele-specific genes/transcripts and splicing events identified from long-read data in preimplantation embryos

This section remains confusing to interpret without having the reciprocal cross.

Fig 5a. vertical axis label = Percentage of what?

Line 431. New paragraph "next we compared the allele specific transcripts identified ..."

Fig 5d. and 5e. It is not clear how the allele-specific analysis was done for short reads. I think this section could be summarized briefly here and the more extensive description moved to supplement.

Line 460. "Among these allele-specific splicing transcripts, a special kind of splicing event (ranging from 0 to 51) involving the DBA-specific and C57-specific mRNA adaptations was observed"

Response to reviewer #1:

The authors have gone to very significant lengths to address all reviewer concerns. I am satisfied with the revised manuscript and think it is a solid piece of work that will be appreciated by the research community.

Response: Thanks very much for the reviewer's for the recognition of our work.

Response to reviewer #2:

1. I agree with the fellow reviewer that the H3K4me3 enrichment at novel TSSs is very weak and not very convincing implying the discovery of false-positives. A comparison with CAGE data would be most helpful.

Response: Thanks very much for the reviewer's constructive suggestions. We quite agree with the reviewer that CAGE data will be helpful for validating novel transcripts at novel TSSs. We have to apologize for not able to perform CAGE experiment currently. In the specific period of 2019-nCoV in China, the students cannot come back to laboratory and the laboratory as well as mice house are totally closed.

Fortunately, we checked the CAGE-containing dataset and found the CAGE data in mouse ESCs is available. Scientifically, ESC stage is close to blastocyst stage *in vivo*. To further verify our identified novel TSSs, we collected public data sets of CAGE peaks from FANTOM5 ¹ and a CAGE data from a previous study regarding ESCs ². Then we compared the results from ESC CAGE or FANTOM5 plus ESC CAGE (FANTOM5+ESC) with our identified merged high-confidence novel transcripts from all seven stages. Moreover, we also directly compared the ESC CAGE data with high-confidence novel transcripts discovered from blastocyst. The annotated genes and novel transcripts with classic TSSs served as positive control.

We defined the promoters of the high-confidence TSSs-linked genes as the regions of 500 bp upstream and downstream of these TSSs, and we compared these promoters with the CAGE peaks identified in ESC CAGE or FANTOM5+ESC CAGE. The data shown below demonstrated as below:

[Redacted]

We agree with the reviewer that H3K4me3 enrichment at all novel TSSs-related genes is a little weak, possibly including some false positive transcripts, so we moved this figure into Supplementary Fig. 5a as a processing data. Then we performed further analysis to identify high-confidence H3K4me3 enrichment, which showed much stronger signals (revised Fig. 2d). To make our conclusion more solid, we present the enrichment of H3K4me3 for high-confidence novel TSSs and novel genes into revised Fig. 2d. When the high-confidence novel TSSs from blastocyst were overlapped with ESC CAGE data, we revealed that over 80% of high-confidence novel TSSs from blastocyst were supported by CAGE peaks.

We sincerely apologize for not able to perform CAGE experiment, as suggested by the reviewer, which will be very helpful for supporting our data.

[Redacted]

2. The allele-specific section is improved by being shortened; I would still argue it should be deleted entirely because it is meaningless without the reciprocal cross. This would not take anything away from the overall strength of this manuscript, in particular now that the authors include KO data of two candidate genes.

Response: Thanks very much for the reviewer's constructive concerning about the allele-specific analysis. According to the reviewer's suggestion, we deleted this section in the revised manuscript.

3. Line 132 ff: "...we profiled the transcriptome using both long- and short-read sequencing across mouse preimplantation developmental stages in female C57BL/6J and male DBA/2 inbred mice". This presumably is meant to read, "in embryos derived from C57BL/6J (female) x DBA/2 (male) crosses." Please correct.

Response: Thanks for the reviewer's kind reminding. We have revised sentence into "we profiled the transcriptome using both long- and short-read sequencing across mouse preimplantation developmental embryos derived from C57BL/6J (female) x DBA/2 (male) crosses".

4. Lines 157-159, it is still unclear what ""sequential subset of exons contained within annotation" means, and how it is different from novel isoform. A diagrammatic depiction of these categories in the supplement is needed.

Response: Thanks for the reviewer's suggestions. We supplied a new diagrammatic depiction to explain the different categories as shown in Figure 1a to provide a more comprehensive outline of our analysis pipeline.

Figure R2. A diagrammatic depiction of long-read transcripts of different categories (also shown in Figure 1a).

5. Line 184: only 54.7% of novel [pls insert] transcripts

Response: Thanks very much for the reviewer's kind reminding. We have revised mentioned phrase into "only 54.7% of novel transcripts".

6.Lines 250-253: "Among 6,289 potentially novel isoforms, 4,273 transcripts possessed classic TSSs overlapping with the TSSs of previously annotated isoforms, while 2,014 novel isoforms [not TSS] possessed potentially novel TSSs; among the potentially novel genes, 2,266 transcripts might have potentially novel TSSs. Please change TSS to novel transcripts (if my interpretation of this sentence is correct).

Response: Thanks for the reviewer's helpful reminding. We are so sorry for making a mistake. We have revised mentioned phrase into "2,014 novel transcripts".

7. Line 297: It is not the case that "thousands" of novel genes and transcript isoforms were "verified". Only some 10 were verified. Please rephrase.

Response: Thanks for the reviewer's kind reminding. We have revised the mentioned sentence into "Thus, we identified a large number of novel long-read transcripts, including thousands of potentially high-confidence novel isoforms and genes, and improved the annotation of preimplantation transcriptome."

8. Lines 344ff and 350ff: Too many confusing abbreviations, please delete SF and DAS and leave the words spelled out.

Response: Thanks for the reviewer's kind reminding. We have revised SF into "splicing factor" and DAS into "differential alternative splicing".

9. Fig. 4G, please provide the numbers of embryos scored. The significance values appear too marginal for the large discrepancies of the bars; please also provide primary data of these counts.

Response: Thanks very much for the reviewer's concerns. We have added the total number of embryos scored in Fig. 4g, as control (5/46), Kdm4dl KO (28/45), and (28/41) from two independent replicates (abnormal/total in each group). The two independent replicates were control (0/16; 5/30), Kdm4dl KO (16/24; 12/21), and (16/21; 12/20). The significance values appear marginal for the large discrepancies of the bars is because of 0 abnormal embryos in one of a control group (0/16, as shown in Fig. 4e).

Response to reviewer #3:

This manuscript describes long-read sequencing of the transcriptome of the pre-implantation embryo. The manuscript has been improved with this revision, but further work is required to make it more easily understandable. The technical description of the transcriptome dataset is extensive and the data is likely to be a useful resource for studies of the pre-implantation embryo. The novelty of the data is hard to assess as there is currently no comparison with the state-of-the-art knowledge of the early embryonic transcriptome. The manuscript could be improved with more exploration or demonstration of the biological significance of the findings.

Fig 1. High-resolution annotation of the preimplantation mouse embryo transcriptome with a combination of long- and short-read sequencing

1. Overall in this section I'd be interested in how the data compares with the state of the art knowledge of pre-implantation embryo transcriptomes. This was raised by another reviewer and was addressed in reviewers response, but there is still no description in the main text.

Response: Thanks very much for the reviewer's helpful comments. In summary, we have made following contributions to scientific community:

As agreed with the reviewer, we have provided an extensive dataset which is useful resource for studies of the pre-implantation embryo. (a) Taking Kdm4dl as an example, our long-read data help us to correctly define the real coding frame of pseudogenes or uncertain genes. (b) We can also increase our recognition of splicing isoforms for a specific gene, and help to accurately identify the specific essential isoform of a gene in early embryogenesis. (c) We also largely increased the transcripts pool for early embryogenesis. For example, according to the expression of novel genes, we identified XLOC_004958 as an essential noncoding gene for normal embryo development.

We agree with the reviewer that it will be more significant if we can compare with the state-of-the-art knowledge of the early embryonic transcriptome. Actually, we have made

partial comparison between annotated transcriptome and our augmented transcriptome. The expression (short-read data) of GENCODE-annotated transcripts possessing potential novel splicing isoforms (from long-seq data) was compared by using GENCODE annotation and augmented annotation as reference separately. δ represents the absolute difference value for a transcript from two annotations. Bar plot showing the proportions of transcripts changed ($\delta > 5$) and transcripts not changed ($\delta \leq 5$). As shown in Supplementary Fig. 5f, the distribution of differential transcription was presented (the yellow part within the circos plot), and those transcripts were statistically analyzed according to their reads number changes (reads mapping by using GENCODE annotation or augmented annotation), showing that about 20% of these transcripts were significantly changed ($\delta > 5$) (Fig. 2f). It demonstrated that with our augmented annotation by long-read data, the expression of transcripts with novel splicing forms will be more accurate than GENCODE annotation. In addition, similar to annotated transcripts (Supplementary Fig. 5g), the novel transcripts identified in the present study showed stage-specific and obvious transcription (Supplementary Fig. 5h).

According to the reviewer 2's and 3's suggestions, we have made comparison of our merged transcriptome (long-read+short-read) for oocyte with the published oocyte transcriptome. In our revision, we have added these data into Supplementary Fig. 2e-f to compare our data with the state-of-the-art knowledge of the oocyte transcriptome. We described this part as "Taking oocyte for an example, we compared our merged transcriptome for oocyte stage with a published oocyte transcriptome³ by short-read sequencing. It showed that the number and length of exons in proportion from our merged transcripts were comparable with the published oocyte data³ (Supplementary Fig. 2e-f), further verifying the reliability of our augmented transcriptome".

Consistent with the reviewer's view, we have made efforts to explore the biological significance of the findings. We encountered the following obstacles: (1) for novel genes with coding potential, they were totally new and we can just postulate their functions according the sequence conservation (Supplementary Fig. 4a); (2) a large percentage of novel transcripts belongs to non-coding genes, and the functions of non-coding RNAs remains to be elucidated; (3) for annotated regions, the novel transcripts identified here (mainly belongs to novel splicing isoforms), which have not been identified before, were not in conserved patterns to be found by long-read data. Therefore, we summarized that we have augmented a more precise transcriptome with augmented transcripts for early embryogenesis, which will help scientists to explore their focused field or process, such as ZGA for us. Our data indeed help us to identify *Kdm4dl* and *XLOC_004958* functionally involved in early embryo development.

2. Figure 1A is helpful, but does not seem to precisely match the workflow as described in the text.

Response: Thanks very much for the reviewer's helpful comments. We revised our Figure 1A to precisely describe the workflow in our study. In the revised Figure 1A, we

clearly described where the long-read and short-read sequencing data come from. We also added a more detailed diagrammatic depiction of annotation of transcripts. Furthermore, the part related to allelic-specific transcripts is removed according to Reviewers' helpful suggestions.

3. Line 138-146. The batching strategy remains unclear.

4. Line 138. Should this be: “For each stage Total RNA samples were precipitated...”?

5. Line 142. The samples subject to short read sequencing, and the purpose of short read sequencing is not clearly stated.

Response: Thanks very much for the reviewer's helpful comments. We are so sorry for our confusing description. We rephrased as following sentences, and the purpose of short read sequencing was also stated.

“Two batches of samples at each stage were collected. Pooled embryos at each stage from one batch, including 150 oocytes (Oo), 150 1-cell embryos (1C), 100 2-cell embryos (2C), 50 4-cell embryos (4C), 25 8-cell embryos (8C), 20 blastocysts (BL, 32-64C), and bulk sperms, were collected for cell lysis. For each stage, total RNA samples were precipitated with absolute ethanol, and the dissolved RNAs were reverse-transcribed and amplified as previously described⁴. Then, the amplified cDNAs were purified. Two batches of cDNAs at each stage were mixed together in equal amounts to obtain a sufficient amount of cDNAs (up to 2 µg) for library construction and subsequent sequencing on the Pacific Biosciences (PacBio) SMRT platform. To compare with long-read sequencing parallelly, a small fraction of the purified cDNAs of each batch were also subjected to short-read sequencing.”

In order to make it clearer, we provided a diagram to present our experimental strategy as below.

6. Fig 1B and C. It is confusing to use two different plot formats to show the same information.

Response: Thanks very much for the reviewer’s concerns. We are so sorry for our unclear description. In Fig. 1b, we presented the annotation of transcripts identified in seven stages specifically. In Fig. 1c, we presented the annotation of the transcripts, which were the merged transcripts from seven stages (deduction of overlapping transcripts between differential stages). To better distinguish the results in Fig. 1c from Fig 1b, we revised the figure legend of Fig. 1c as “Annotation of merged long-read transcripts that were the combination of transcripts identified across seven stages. The numbers and percentages of merged transcripts in the five categories are presented.”

7. Line 169. Should this be: “revealing 2,280 transcripts from potential novel genes”?

Response: Thanks very much for the reviewer’s kind reminding. We have corrected the mentioned sentence in revision.

8. Line 171. What is the number of full length PacBio reads supporting novel

transcripts / genes compared with annotated transcripts? This would also be good validation to have. Fig 1D/E How do these categories align with those in figure C? Does novel transcripts include both novel gene and novel isoform?

Response: Thanks very much for the reviewer's helpful comments. We added a bar plot similar to Fig. 1f to show the number of full length PacBio reads of annotated transcripts and novel transcripts (including novel isoforms and transcripts from novel genes). Due to the low expression of these novel transcripts, the number of full length PacBio reads supporting novel transcripts is smaller than that supporting annotated transcripts.

As we defined in Fig. 1b, the annotated transcripts contained transcripts that were exactly matched to the annotation or transcripts contained within the GENCODE annotation. And novel transcripts contained "potentially novel genes" (no overlapping with the reference annotation) and "potentially novel isoforms". To make it clearer, we revised the first sentence describing Fig. 1d/e as the following sentence to interpret the definition of novel transcripts. "Next, the merged transcripts, including the annotated transcripts (transcripts that were exact match to annotation and sequential subset of exons of transcripts that were contained within annotation) and the novel transcripts (potentially novel isoforms and novel genes)"

[Redacted]

9. Line 189. Can you speculate why the fraction of coding vs non-coding is so different for annotated vs novel transcripts? Perhaps the comparison should be done against a more comprehensive reference annotation database (e.g. including <http://www.noncode.org/>)

Response: Thanks very much for the reviewer's helpful concerns. To assess the fraction of non-coding transcripts in annotated transcripts and novel transcripts, we used the combination of GENCODE and NONCODE database as reference annotation. Then we annotated the transcripts merged from seven stages using the combination of GENCODE

and NONCODE. [Redacted]

10. Line 192. Here we are introduced to transcripts identified from short-read data, but there is no description in the main text (or in fig 1A) of where these come from.

Response: Thanks very much for the reviewer's kind reminding. We have revised the Fig. 1a to provide a clearer outline of the long-read and short-read data analysis, and we also added a more detailed description of the short-read data used in the main text related to Fig. 1a as "To compare with long-read sequencing parallelly, a small fraction of the purified cDNAs of each batch was also subjected to short-read sequencing (Fig. 1a)."

11. Fig 1f. As for fig1 d,e, what is the definition of novel transcript? It could be that much of the short read mapping signal may be coming from regions that are shared with known transcripts.

Response: Thanks very much for the reviewer's concerns. We defined the novel transcripts as the combination of "potentially novel genes" and "potentially novel isoforms". The merged transcripts described in Fig. 1c were applied in Fig. 1f. To quantify the TPM of novel transcripts, we used salmon in quasi mode to assign reads to transcript uniquely and calculated the TPMs of transcripts. In this mode, each reads will only be counted one time as the belonging to one transcript, even though the reads may come from an exon shared by two transcripts. This method can avoid the condition in which the short-reads originated from known transcripts were assigned to novel transcripts to the largest extent.

According to this method, potentially novel genes were recognized from unannotated regions, which cannot be overlapped with known transcripts. For potentially novel isoforms, there were only two kinds of transcripts defined as novel isoforms: there are new coding regions or exons (at the 5'-end, middle, or 3'-end) compared with known transcripts; there were missing coding regions or exons in the middle of known transcripts. For those short-length transcripts derived from known transcripts, if they are totally matched within known transcripts, these transcripts are defined as "Sequential subset of exons contained within annotation". Accordingly, this method can guarantee to avoid assigning "Sequential subset of exons contained within annotation" to novel transcripts to the largest extent.

12. Lines 190-213. The comparison with short read data could be described more succinctly.

Response: Thanks very much for the reviewer's helpful suggestions. We have shortened the first paragraph that compared the transcripts assembled from long-read data and short-read data as below. The detailed comparison method was added in supplementary methods.

"Moreover, we compared the novel transcripts identified from long-read data and/or short-read data with the annotated transcripts from GENCODE (Supplementary Fig. 3; Supplementary Methods). In general, novel transcripts identified from the long-read data were more similar to the GENCODE annotation than those from the short-read data (Supplementary Fig. 3a-f). In addition, the transcripts that could only be identified from long-read data displayed slightly more similar characteristics to the GENCODE-annotated transcripts than the transcripts that could only be identified from short-read data (Supplementary Fig. 3g-l)."

13. Line 217 and Fig 2a. There is no explanation of what short read plus long read actually means. Was this some kind of co-assembly or merging of transcriptomes. How was this done? What about long-read only?

Response: The "short read plus long read" means the combination of transcripts derived from long-read data and short-read data, which was archived by merging the two transcriptomes using Cuffmerge. Due to the much smaller number of long-read only

transcripts, which is the consequence of the technical limitation of the third-generation sequencing, we do not add the curve in Fig. 2a.

To better explain the definition of short-read plus long read, we revised the 217-219 as “To assess the ability of identifying novel transcripts with long-read sequencing data, we compared the saturation curve of the novel transcripts identified from short-read data only with that of novel transcripts derived from merging transcripts identified from short- and long-read data.”

The figure legend of Fig. 2a was revised as “Saturation of novel transcripts identified by short-read sequencing and via a combination of short- and long-read sequencing. This combination was achieved by merging transcripts identified from short- and long-read data. We merged the identified novel transcripts from each stage with short-read sequencing data and a combination of short- and long-read sequencing data step by step in 100 random orders. The identified novel transcripts were annotated with the GENCODE annotation at each step. The lines and bands represent the mean and the 99% confidence interval of the number of novel transcripts identified at each step, respectively.”

14. Line 235. New paragraph for section on non-coding sequences

Response: Thanks very much for the reviewer’s kind advice, and we have corrected it in the revised manuscript.

15. Line 238, Fig 2C. Is there any complementary analysis of signatures of lncRNA that can be done here.

Response: Thanks very much for the reviewer’s concerns. As shown in Fig. 1e, 94.5% of novel non-coding transcripts were identified as lncRNAs. We also analyzed the conservation of novel lncRNA only [Redacted], showing highly similar results (1,168 lncRNAs showed higher conservation scores; 1,183 transcripts among the novel noncoding transcripts showed higher conservation scores (Fig. 2c)) with all novel non-coding transcripts. Thus, we remain to present this part as shown in Fig. 2c.

[Redacted]

16. Line 263. New paragraph "To identify high-confidence.....".

Response: Thanks very much for the reviewer's kind advice, and we have corrected it in the revised manuscript.

17. What is the coding / non-coding breakdown of the high confidence novel TSSs.

Response: Thanks very much for the reviewer's suggestions. To assess the coding potential of the transcripts which possessed the high confidence novel TSSs, we applied the same analysis pipeline using CPAT.

[Redacted]

18. Line 284-296 and Fig 2F. I don't understand the analysis or the figure.

Response: Thanks very much for the reviewer's concerns. To show how we can take advantage of the augmented annotation by integrating our long-read data with GENCODE annotation, we compared the difference between mapping results using two different annotation (GENCODE vs. GENCODE+long-read). The transcriptome annotation can contribute to the accuracy of the alignment in RNA-seq analysis. So we mapped the short-read data to mm10 with GENCODE-annotation and our high-resolution annotation. By comparing the number of reads mapping to the shared transcripts between two transcriptomes, we found that using high-resolution transcriptome can make a significantly difference in quantifying the expression level of transcripts with novel identified splicing isoforms. To make it easier to be understood, we moved the Circos plot into Supplementary Fig. 5f. We also performed analysis with a bar plot to illustrate the difference of the expression levels of transcripts when using two annotations. On average, about 20% of transcripts in all transcripts possessing novel splicing isoforms in our augmented annotation were quantified differently with ($\delta > 5$, calculated from the difference value of reads number between GENCODE vs. GENCODE+long-read) when using two different annotations (Fig. 2f). It was worthy noticing that only transcripts possessing novel splicing isoforms in our augmented annotation were statistically analyzed here, because the annotation of novel genes will not change the expression of known GENCODE annotated genes. For example, a gene harbors 2 isoforms in GENCODE, while we identified a novel splicing isoform for this gene in long-read data. Then, the RNA-seq reads will be mapped to 3 transcripts but not 2 transcripts. In circle plot, only the original 2 isoforms will be recalculated by using augmented annotation (deduction of the expression reads of the novel isoform), and the difference values for the original 2 isoforms between GENCODE vs. GENCODE+long-read were presented.

Fig. 2f The expression of GENCODE-annotated transcripts possessing potential novel splicing isoforms (from long-seq data) was compared by using GENCODE annotation and augmented annotation as reference separately. δ represents the absolute difference value for a transcript from two annotations. Bar plot showed the proportions of transcripts changed ($\delta > 5$) and transcripts not changed ($\delta \leq 5$).

To improve the readability of this section, we revised the paragraph as follow. “By combining these novel transcripts, we augmented the GENCODE annotation with the novel transcripts derived from merged transcripts. To assess how different transcriptome annotation can affect the quantification of transcripts, we quantified the mapping reads of the GENCODE-annotated transcriptome and augmented transcriptome using salmon (v0.10.0) ⁵. We observed that for GENCODE-annotated transcripts possessing potential novel splicing isoforms, about 20% of these transcripts showed large mapping reads changes ($\delta > 5$) for the newly augmented transcriptome references when using the augmented annotation as reference ($P < 1 \times 10^{-100}$, Wilcoxon matched-pairs signed rank test; **Fig. 2f and Supplementary Fig. 5f**).”

We also added more detailed figure legend of the Circos plot as following: “Circos plot showing the expression levels of differentially expressed transcripts ($\delta > 5$) annotated when mapping short-read data to genome with GENCODE annotation and GENCODE-augmented annotation as reference separately. Red: the expression level of differentially expressed transcripts based on the GENCODE reference. Blue: the expression level of differentially expressed transcripts based on the augmented reference (combination of long-read sequencing and GENCODE). Yellow: the altered expression of differentially expressed transcripts annotated according to the two references.”

19. Line 298. The statement “we.... improved the annotation of preimplantation transcriptome” suggests a comparison with prior knowledge of the preimplantation transcriptome. This was not done.

Response: Thanks very much for the reviewer’s concerns. The section describing **Fig. 2f and Supplementary Fig. 5f** meant to illustrate the improvement of the annotation of transcriptome compared with the GENCODE annotation. In the revised manuscript, we

compared our transcriptome with the previous transcriptome of oocyte as the previous comments suggested (Supplementary Fig. 2e-f).

20. Lines 333-339 and Fig 3d. I'm not sure I understand this. Does 'gain' and 'loss' refer to presence / absence of a transcript isoform? Or is this a quantitative comparison (e.g. 73% transcript isoforms at the 2C stage were more highly expressed than at the 1C stage). Either way this needs clarifying.

Response: Thanks very much for the reviewer's concerns. The "gain" and "loss" in Fig 3d refer to the presence and absence of transcripts between consecutive stages based on the identification of transcripts in each stage. To clarify the definition of "gain" and "loss", we added following sentence as following. "By comparing the transcriptome in a specific stage with its previous stage, we identified gained transcripts that were absent in a previous stage while present in a current stage, as well as lost transcripts that were present in a previous stage while absent in a current stage. The percentages of gained and lost transcripts were balanced in the majority of consecutive stages, with an average of 45.9% gained and 46.8% lost transcripts, respectively."

21. Line 343. New paragraph "To explore the potential factors....."

Response: Thanks very much for the reviewer's kind advice, and we have corrected it in the revised manuscript.

22. Lines 351-353. This is a bit tangential but also interesting. Does the correlation between SF expression and alternative splicing make sense in the context of the known modes of action of any of the SFs?

Response: Thanks for your concerning about the splicing factor analysis. A previous study which analyzed the correlation between the splicing factor expression and the frequency of skipped exon, inspired us to analyze the underlying relationship between splicing factors and alternative splicing events in preimplantation embryo⁶. In our study, we profiled the dynamic changes of splicing events during the preimplantation embryo. By assessing the correlation between the stage-specific expressed splicing factors and the number of splicing events during the preimplantation stages, we found that some stage-specific expressed splicing factors were positively or negatively correlated with splicing events. It indicates that these splicing factors may directly involve in the dynamic changes of splicing events.

23. Fig 3e. How does this relate to fig 3d? Does 'down' mean the same thing as 'loss', and up the same as 'gain'? In general this plot is hard to interpret.

Response: Thanks very much for the reviewer's concerns. In Fig. 3d, the gained transcripts referred to the transcripts absent in previous stage while present in current stage. In contrast, the lost transcripts were defined to be the transcripts that were present in previous stage while absent in current stage.

In Fig. 3e, the "up" and "down" was analyzed to identify the differential alternative splicing events. During splicing step, a pre-mRNA can produce one or more functional transcripts through a process called alternative splicing (AS). When an alternative splicing event was compared at two consecutive stages, the proportion of two spliced isoforms were significantly changed, then this event is defined as a differential alternative splicing event. The percent spliced in index (PSI) used here indicates the fraction of the total transcript amount from each locus accounted for by a specific splicing event. Alternative splicing event with PSI value significantly changed between two stages were defined as differential alternative splicing events. Here the "up" refers to $\Delta \text{PSI} > 0$, $P < 0.05$, which means the fraction of the alternative exon up-regulated. The "down" refers to $\Delta \text{PSI} < 0$, $P < 0.05$, which means the fraction of the alternative exon down-regulated. To make the river plot clearer, we replaced the "up" and "down" with "PSI upregulation" and "PSI downregulation" in Fig 3e, respectively.

24. Line 365. I don't understand the sentence "Intriguingly, rare significant DAS events were retained from the 1-cell to blastocyst transitions".

It might be useful to show genome browser shots of novel DAS for a few 'famous' embryonic development genes.

Response: Thanks very much for the reviewer's suggestions. In Fig. 3e, we profiled the dynamic changes of DAS events during the preimplantation stages using river plot. Most of the DAS events occurred at the specific time point during the transition, which means that most DAS events were detected at specific stage transitions. The specificity of DAS events indicated that few transcripts were continuously regulated. The reviewer's suggestion of providing a genome browser shot of novel DAS of genes was very precious.
[Redacted]

25. Line 372. New paragraph “To identify the potential novel transcripts involved in ZGA....”

Response: Thanks very much for the reviewer’s kind advice, and we have corrected it in the revised manuscript.

Fig 4. Identification of two novel transcripts functionally involved in early embryogenesis

26. What are the exact genomic coordinates of the regions deleted?

Response: Thanks very much for the reviewer’s concerns. The genomic coordinates information was added in the Fig. 4d. The sgRNA sequences were also included in Supplementary Table 13.

Fig 5. Allele-specific genes/transcripts and splicing events identified from long-read data in preimplantation embryos

27. This section remains confusing to interpret without having the reciprocal cross.

28. Fig 5a. vertical axis label = Percentage of what?

29. Line 431. New paragraph “next we compared the allele specific transcripts identified”

30. Fig 5d. and 5e. It is not clear how the allele-specific analysis was done for short reads. I think this section could be summarized briefly here and the more extensive description moved to supplement.

31. Line 460. “Among these allele-specific splicing transcripts, a special kind of splicing event (ranging from 0 to 51) involving the DBA-specific and C57-specific mRNA adaptations was observed”

Response: Thanks very much for the reviewer’s comments on Fig. 5. According to the reviewer 2’s and reviewer 3’s suggestions, we have deleted the entire part regarding allele-specific splicing events in our revision.

We also give a simple response to the reviewer’s question as shown below:

(28) The mentioned y-axis label should be “Percentage of allelic-specific transcripts” in Fig 5a.

(30) Short-read data-based allele-specific analysis: N-masked genome was constructed based on the DBA/2-specific SNPs and used as the mapping reference for the following analysis. Short-read data were mapping to the N-masked genome using STAR. Uniquely mapped RNA-seq reads were split into a DBA/2-specific group, a C57/BL6J-specific group and an unassignable group by using SNPsplit based on the nucleobases of N-sites. The allelic expression levels (FPKM) of genes and isoforms were calculated by Stringtie using parental-specific reads as the input. The allelic ratio was calculated as $\frac{\text{Reads}_{\text{C57}}}{\text{Reads}_{\text{C57}} + \text{Reads}_{\text{DBA}}}$. Allelically-imbalanced (allelic ratio < 0.33 or allelic ratio > 0.66) and allelically-balanced (allelic ratio >= 0.33 and allelic ratio <= 0.66) transcripts were defined according to their allelic ratios.

References for response letter

1. Lizio M, *et al.* Gateways to the FANTOM5 promoter level mammalian expression atlas. *Genome Biol* **16**, 22 (2015).
2. Lloret-Llinares M, *et al.* The RNA exosome contributes to gene expression regulation during stem cell differentiation. *Nucleic Acids Res* **46**, 11502–11513 (2018).
3. Gahurova L, *et al.* Transcription and chromatin determinants of de novo DNA methylation timing in oocytes. *Epigenetics Chromatin* **10**, 25 (2017).
4. Chen J, Suo S, Tam PP, Han JJ, Peng G, Jing N. Spatial transcriptomic analysis of cryosectioned tissue samples with Geo-seq. *Nat Protoc* **12**, 566–580 (2017).
5. Patro R, Duggal G, Love MI, Irizarry RA, Kingsford C. Salmon provides fast and bias-aware quantification of transcript expression. *Nat Methods* **14**, 417–419 (2017).
6. Chen H, *et al.* Long-Read RNA Sequencing Identifies Alternative Splice Variants in Hepatocellular Carcinoma and Tumor-Specific Isoforms. *Hepatology* **70**, 1011–1025 (2019).

Reviewers' Comments:

Reviewer #2:

Remarks to the Author:

The authors have made a substantial effort to address all points raised. Even if not every detail is 100% satisfactory and/or convincing and some of the "novel" genes/isoforms may prove not to be correct, this analysis is a very valuable resource and study for the field, highlighting the need for stage- and tissue-specific re-annotations of the reference genome(s) for a more accurate analysis of common RNA-seq data. I would also like to acknowledge the authors' resourceful use of deposited CAGE data in the absence of being able to perform these experiments directly during challenging times.

Reviewer #3:

Remarks to the Author:

The authors have made a good effort at addressing all of my comments.

I'm concerned that the 'novel' transcripts have such different properties to known transcripts. Compared with known transcripts these are: 1) less likely to be protein coding, 2) less likely to be marked by H3K4Me3 at TSS, 3) less likely to overlap a CAGE tag in ES cell data and 4) supported by fewer full length reads (Figure R3). Together this makes me suspicious that many of these may be spurious transcripts of no biological function. You may want to quote the number of high confidence novel transcripts based on some filtering criteria (rather than all 2,280 as currently stated in the abstract). It would also be useful to provide a single large supplemental table with rich metadata annotation about each transcript (Gene name, known gene name, coordinates, length, #exons, number long reads per condition, CAGE tag, H3K4Me3, etc, etc) so that readers can make their own calls about high quality transcripts. I couldn't easily find this in existing supplementary tables.

I would find this paper much easier to read if there was less emphasis on counts of transcripts, isoforms and splicing types, and more placing the results in context of biology of early embryonic development.

REVIEWERS' COMMENTS:

Reviewer #3 (Remarks to the Author):

The authors have made a good effort at addressing all of my comments.

I'm concerned that the 'novel' transcripts have such different properties to known transcripts. Compared with known transcripts these are: 1) less likely to be protein coding, 2) less likely to be marked by H3K4Me3 at TSS, 3) less likely to overlap a CAGE tag in ES cell data and 4) supported by fewer full length reads (Figure R3). Together this makes me suspicious that many of these may be spurious transcripts of no biological function. You may want to quote the number of high confidence novel transcripts based on some filtering criteria (rather than all 2,280 as currently stated in the abstract). It would also be useful to provide a single large supplemental table with rich metadata annotation about each transcript (Gene name, known gene name, coordinates, length, #exons, number long reads per condition, CAGE tag, H3K4Me3, etc, etc) so that readers can make their own calls about high quality transcripts. I couldn't easily find this in existing supplementary tables.

I would find this paper much easier to read if there was less emphasis on counts of transcripts, isoforms and splicing types, and more placing the results in context of biology of early embryonic development.

Response: Thanks very much for your concerning about the reliability of the novel transcripts.

(1) Less likely to be protein coding:

In our prediction of protein coding ability of novel transcripts, we have some difficulties: (a) these novel transcripts are totally new and their protein coding ability can be only predicted according to software calculation; (b) a large percentage of novel transcripts belongs to non-coding RNAs; (c) previous studies gave more focus on coding genes, and the number of novel splicing isoforms that mainly belong to novel splicing isoforms is relatively small. Relatively, the studies on non-coding RNAs became popular in recent years.

(2) Less likely to be marked by H3K4Me3 at TSS

We agree with the reviewer that H3K4me3 enrichment at all novel TSSs-related genes (n=2280) is a little weak, so we performed further analysis to identify high-confidence H3K4me3 enrichment, which showed much stronger signals (n=1007; revised Fig. 2d). Even for those 1273 (n=2280-1007) transcripts, 472 novel genes possess two or more exons, the majority of which possess classical GT/AG splicing junctions, and these transcripts should be

also confident novel genes according to splicing rules. For those novel genes with weak H3K4me3 enrichment, it is possible that these genes are expressed with low but detectable expression levels or some novel genes might be maternally remained (not relevant to H3K4me3-activated transcription).

(3) Less likely to overlap a CAGE tag in ES cell data

[Redacted]

(4) Supported by fewer full length reads:

During the analysis of the long-read data, the expression levels were not considered because of the limitations of the technology. Especially for those short transcripts with low mRNA copies, even with only 1 copy, as long as they were successfully ligated into libraries, the short transcripts can generate multiple copies of reads by circling sequencing of a long reads, and these lowly-expressed transcripts can be captured by the third-generation sequencing technology with not that low reads (in another words, the sequencing reads from long-read data were not 100% matched with real copies and short-read data), while these transcripts might be missed or designated with very low reads in short-read data. As we illustrated in our results, the novel transcripts, especially the novel transcripts from unannotated loci, were usually expressed with a relatively low level.

(5) The number of novel genes (n=2280) and supplemental table with rich metadata annotation.

For those 1273 (n=2280-1007) transcripts that are not accord with our criteria of high-confidence transcripts, 472 novel genes possess two or more exons, the majority of which possess classical GT/AG splicing junctions, and these transcripts should be also confident novel genes according to splicing rules. Therefore, we cannot exclude those 1273 transcripts as novel genes. We agree with the reviewer about the inaccurate description about 2280 novel genes. We revised the description as “2280 potential novel genes” in revised abstract and main text. We also added a sentence to describe this concern as “Besides, for those potential novel genes not accord with our high-confidence criteria, 472 transcripts possessed two or more exons.” We also According to the reviewer’s helpful suggestions, we have added a new supplementary table with rich metadata annotation as supplementary data 4.

(6) According to the reviewer's helpful suggestions, we have deleted some unnecessary descriptions with exact numbers. Considering the advantage of long-read sequencing for augment of transcriptome annotation, we hope to keep some necessary numbers regrading novel transcripts and genes.

We agree with the reviewer that it will be more meaningful with more biological exploration of early embryogenesis, and we have performed some analysis to discuss the relationship between alternative splicing and embryogenesis. We also elucidate the functions of the novel isoform of *Kdm4dl* and the novel gene *XLOC_004958* in early embryogenesis. Because of the low throughput in functional assays, we are not able to validate all identified novel transcripts, and we will perform some screening assays in our

future study to explore their functions in detail. Here we added some discussion about this concern in revision. Furthermore, with the constructive suggestions from the reviewer 3, we will try to explore the parent-of-origin effects with backcrossing strategy, to fully dissect the biological significance during this process in our future study.